# LAW & ORDER: Adaptive Spatial Weighting for Medical Diffusion and Segmentation

**Anugunj Naman** *anaman@purdue.edu*
*Elmore Family School of Electrical and Computer Engineering*
*Purdue University, West Lafayette, USA*

**Ayushman Singh** *ayushman@sesame.com*
*Sesame AI, New York, USA*

**Gaibo Zhang** *zhan5117@purdue.edu*
*Department of Computer Science*
*Purdue University, West Lafayette, USA*

**Yaguang Zhang** *zhan1472@purdue.edu*
*Department of Agricultural and Biological Engineering (ABE)*
*Department of Agricultural Sciences Education and Communication (ASEC)*
*Purdue University, West Lafayette, USA*

**Reviewed on OpenReview:** *https://openreview.net/forum?id=sJXqzr3oLl*

## Abstract

Medical image analysis depends on accurate segmentation and controllable synthesis, but both tasks face severe spatial imbalance: lesions occupy small regions against large backgrounds. We study adaptive spatial weighting as a task-level design principle and instantiate it in two adapters. LAW learns per-pixel loss weights for mask-conditioned diffusion by modulating a ratio prior with a feature-dependent delta map, with normalization, clamping, and Dice regularization for stability. ORDER improves lightweight segmentation by adding selective bidirectional skip attention with stage-wise confidence gating. On held-out diffusion test sets, LAW lowers FID from 158.13±0.15 to 108.43±0.71 on Polyps, from 144.13±0.31 to 89.51±0.96 on KiTS19, and from 139.22±0.38 to 112.58±0.68 on BRISC, while improving held-out mask-recovery Dice from 0.681±0.013 to 0.825±0.003 on Polyps. When the resulting images are added to nnUNet training, downstream Polyps mDice rises from 71.7±0.4 to 74.1±0.8. On the cleaned Polyps segmentation protocol, the reported ORDER configuration reaches 76.3±1.9 mDice and 67.2±2.0 mIoU at 42K parameters and 0.11 GFLOPs, versus 70.3±1.5 mDice and 59.9±1.7 mIoU for matched MK-UNet. On BRISC under the same training recipe, ORDER reaches 77.4±0.8 mDice and 68.1±0.7 mIoU. These results position adaptive spatial weighting as a practical design idea for both medical diffusion and efficient segmentation.

## 1 Introduction

Medical image segmentation and synthesis are essential for diagnosis, treatment planning, and data augmentation. Accurate segmentation localizes lesions precisely (Isensee et al., 2021; Chen et al., 2024; Wei et al., 2021), while controllable synthesis generates diverse training data to address annotation scarcity (Ho et al., 2020; Song et al., 2021b; Rombach et al., 2022; Shorten & Khoshgoftaar, 2019). ControlNet-style conditioning improves spatial control in diffusion models (Zhang et al., 2023; Mou et al., 2024), and lightweight U-Nets improve the efficiency of segmentation backbones (Tang et al., 2024; Ruan et al., 2023).

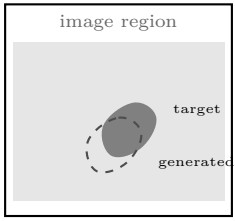

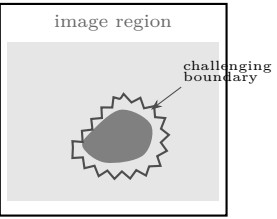

Lesion: ∼7% of pixels
Background dominates loss

Uniform skip fusion
wastes capacity on easy regions

(a) Synthesis: mask drift

(b) Segmentation: boundary-sensitive detail

Figure 1: Spatial imbalance in medical imaging. (a) Lesions occupy a small fraction of pixels, causing uniform loss weighting to prioritize background and drift from target masks. (b) Uniform skip connections fail to adapt their fusion strength across easy and hard regions, especially near lesion boundaries.

Unfortunately, both approaches face a common challenge: **spatial imbalance** (Fig. 1). In medical images, lesions occupy small, irregular regions against vast backgrounds. This imbalance causes diffusion models to drift from prescribed masks during synthesis (Wu et al., 2023; Du et al., 2023), as illustrated in Fig. 1a, and prevents efficient segmenters from adapting skip fusion strength to the spatially difficult regions that dominate their errors (Fan et al., 2020; Dong et al., 2023), as illustrated in Fig. 1b.

Prior work addresses these issues separately. Adaptive diffusion methods reweight losses to emphasize lesions (Du et al., 2023), but use fixed, ratio-based priors that cannot adapt to local feature complexity. Lightweight segmenters reduce parameters by 100× (Ruan et al., 2023; Liu et al., 2024), yet struggle on ambiguous boundaries where heavier models excel (Isensee et al., 2021; Chen et al., 2024). These limitations motivate a shared design lens: allowing the model to learn where additional capacity should be spent under severe spatial imbalance.

**Shared Perspective.** We use **adaptive spatial weighting** as a common perspective for two task-specific adapters, rather than claiming a single formally unified mechanism. In diffusion, the model concentrates denoising effort on lesion regions. In segmentation, the network selectively strengthens late skip interactions when encoder and decoder context indicate that extra refinement is useful. The common thread is to learn where extra modeling emphasis is most useful, even though the resulting modules operate in different architectures and objectives (Fig. 2).

**Our solution.** LAW (Learnable Adaptive Weighter) for mask-conditioned diffusion predicts per-pixel loss modulation from intermediate features and the conditioning mask, building on ratio-based priors but adding learned adjustments stabilized via normalization, clamping, and a Dice regularizer. This yields region-aware denoising under a weighted denoising objective. ORDER (Optimal Region Detection with Efficient Resolution) applies the same idea to segmentation by keeping MK-UNet's multi-kernel encoder (Rahman & Marculescu, 2025) and inserting lightweight bidirectional attention with learned global confidence gates on the final two skips. The bidirectional attention maps are efficiently constructed using shared decoder–encoder similarity, and a learned scalar gate scales both residual updates before fusion, so ORDER concentrates compute on the late skip stages — where semantics and spatial detail must be reconciled — while keeping to only 42K parameters.

## 1.1 Contributions

The contributions of this work are as follows:

- LAW for diffusion extends ratio-based priors with learned, feature-dependent adjustments stabilized by normalization, clamping, and Dice regularization. On held-out test sets it lowers FID from 158.13±0.15 to 108.43±0.71 on Polyps, from 144.13±0.31 to 89.51±0.96 on KiTS19, and from 139.22±0.38 to 112.58±0.68 on BRISC, while improving Polyps mask-recovery Dice from 0.681±0.013 to 0.825±0.003. In downstream augmentation it raises Polyps nnUNet mDice from

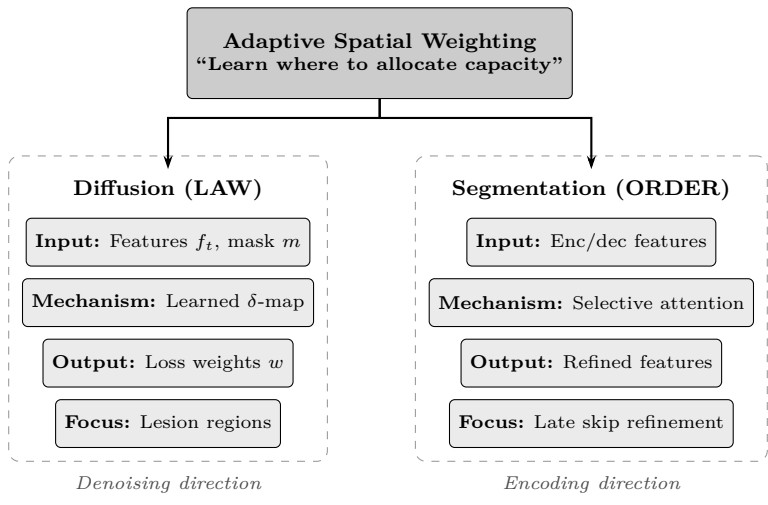

Figure 2: Adaptive spatial weighting as a shared perspective. LAW (for diffusion) and ORDER (for segmentation) both emphasize spatially sparse, difficult regions, while remaining task-specific modules applied at different stages of the imaging pipeline.

71.7±0.4 to 74.1±0.8 and BRISC mDice from 87.7±0.2 to 89.6±0.4, while remaining within 0.1 mDice of the best KiTS19 baseline.

- ORDER for efficient segmentation couples selective bidirectional skip attention with a 42K-parameter backbone. The reported ORDER configuration reaches 76.3±1.9 mean mDice and 67.2±2.0 mean mIoU on the cleaned Polyps protocol at 0.11 GFLOPs, versus 70.3±1.5 mDice and 59.9±1.7 mIoU for matched MK-UNet. On BRISC under the same training recipe, ORDER reaches 77.4±0.8 mDice and 68.1±0.7 mIoU.

- The ablations isolate which components matter. For LAW, learned weighting only helps once paired with normalization, lower/upper clamping, and Dice alignment. For ORDER, the best raw internal Dice occurs at the mixed-depth skips $\{1, 2\}$, while the reported main model uses $\{0, 1\}$ for a better runtime-memory trade-off.

See Fig. 3 for a visual placement of our scope, with the diffusion side summarized in Fig. 3a and the segmentation side in Fig. 3b. The methods build on latent diffusion (Rombach et al., 2022), ControlNet conditioning (Zhang et al., 2023), and efficient U-Net designs (Tang et al., 2024; Ruan et al., 2023), but the adaptive weighting mechanisms are modular with respect to those backbones. The empirical study combines diffusion experiments on Polyps (Ali et al., 2023), KiTS19 (Heller et al., 2021), and BRISC (Fateh et al., 2026) with segmentation experiments on Polyps and BRISC, showing where adaptive spatial weighting is useful and where its downstream benefits are neutral.

## 2 Related Work

Three research threads inform this work: spatial control in diffusion models, efficient segmentation architectures, and adaptive mechanisms for visual learning. Each addresses spatial imbalance differently, but they are usually studied in isolation within either generative or discriminative settings.

### 2.1 Spatial Control in Diffusion Models

Denoising diffusion probabilistic models (Ho et al., 2020; Song et al., 2021b) provide powerful generative frameworks; latent diffusion (Rombach et al., 2022) and accelerated samplers such as DDIM (Song et al.,

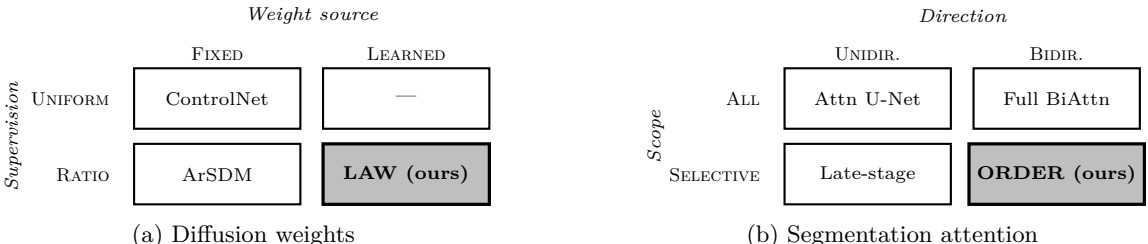

Figure 3: Design space for adaptive spatial mechanisms. (a) LAW combines learned feature-dependent weights with ratio-based priors. (b) ORDER applies bidirectional attention selectively at decoder skip stages. Shaded cells indicate our contributions.

2021a) enable high-resolution, tractable synthesis. In medical imaging they mitigate data scarcity by creating supervisory data while supporting anomaly detection, temporal modeling, and MRI reconstruction (Wu et al., 2023; Nguyen et al., 2023; Qiu et al., 2025a; Rahman et al., 2023; Kim & Ye, 2022; Wolleb et al., 2022; Pinaya et al., 2022; Chung & Ye, 2022).

Spatial control began with ControlNet (Zhang et al., 2023), which copies frozen encoders with zero-initialized connections for mask conditioning. Lighter adapters such as T2I (Mou et al., 2024) and ControlNet++ (Li et al., 2024) further improve efficiency. Yet medical conditioning data is limited and lesions occupy irregular, tiny regions, so background pixels dominate the loss and models drift from the prescribed masks.

Adaptive spatial weighting tackles this drift. ArSDM (Du et al., 2023) reweights the loss with a mask-derived ratio prior, but the fixed heuristic cannot react to local denoising difficulty and can suppress backgrounds excessively. Qiu et al. (Qiu et al., 2025b) train a teacher-student pair instead, raising training cost and tying the weights to the teacher's biases. LAW continues this line by adapting the prior with intermediate features, adding clamping to avoid degeneration, and removing the need for auxiliary pretrained segmenters.

## 2.2 Efficient Segmentation Architectures

Medical segmentation has progressively shifted from heavy encoder-decoders to compact designs. U-Net (Ronneberger et al., 2015) popularized skip-connected decoders, with successors such as U-Net++ (Zhou et al., 2018), V-Net (Milletari et al., 2016), Attention U-Net (Oktay et al., 2020), and transformer hybrids (Chen et al., 2024; Cao et al., 2022; Azad et al., 2022; Heidari et al., 2023) trading accuracy for substantially less computation.

Efficiency-focused architectures draw on mobile design principles (Howard et al., 2017; Sandler et al., 2018). CMUNeXt (Tang et al., 2024) uses large kernels, EGE-UNet (Ruan et al., 2023) leverages group convolutions, and MK-UNet (Rahman & Marculescu, 2025) reaches a low 27K parameters via multi-kernel inverted residual blocks.

Despite these innovations slashing parameter counts by $100\times$, lightweight models often fail on ambiguous boundaries. Uniform skip fusions treat every region identically, so scarce capacity is wasted on easy background pixels instead of lesion edges. ORDER keeps MK-UNet's compact encoder but adds targeted bidirectional skip attention, computing one similarity matrix reused in both directions (Kitaev et al., 2020) and activating it only in the final decoder stages where semantics peak.

## 2.3 Adaptive Mechanisms in Visual Models

Attention and reweighting mechanisms are longstanding tools in vision. Channel/spatial attention (Woo et al., 2018; Oktay et al., 2020; Hu et al., 2018) teaches discriminative models to emphasize salient features, while classifier or classifier-free guidance (Dhariwal & Nichol, 2021; Ho & Salimans, 2021) tune diffusion globally rather than spatially.

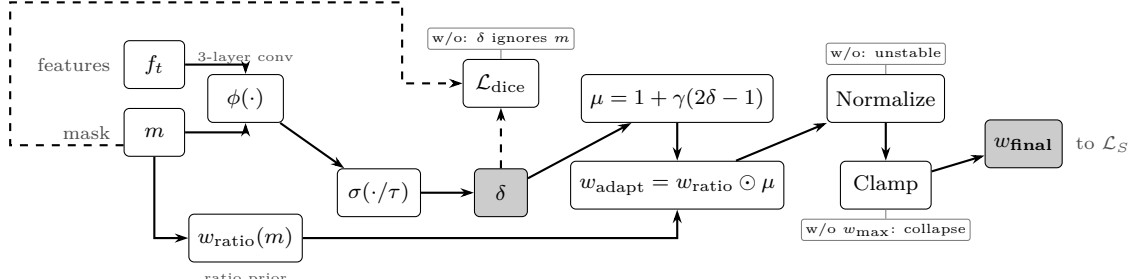

Figure 4: LAW weight computation with stabilization mechanisms. The learned delta map $\delta$ modulates a ratio-based prior, with normalization and clamping ensuring training stability. Annotations indicate failure modes when each component is ablated (cf. Table 6).

Medical synthesis studies extend these ideas with mask-aware loss weighting in ArSDM (Du et al., 2023) and Adaptive Distillation (Qiu et al., 2025b), yet they confine the adaptivity within the diffusion loop. Adaptive foreground emphasis also arises in object-centric representation learning, where Tian et al. (2025) argue that explicit foreground-aware objectives are necessary for slot-based models to recover meaningful objects. The same spatial imbalance appears in segmentation. LAW and ORDER therefore study adaptive weighting in both settings, suggesting that learning *where* to allocate capacity can help both generative fidelity and discriminative accuracy.

## 3 Method

Two instantiations of adaptive spatial weighting are presented: LAW (Learnable Adaptive Weighter) for mask-conditioned diffusion (Section 3.1) and ORDER (Optimal Region Detection with Efficient Resolution) for efficient segmentation (Section 3.2). Both share the principle of learning where to allocate computational resources, but operate in complementary settings.

### 3.1 LAW: Learnable Adaptive Weighting for Diffusion

LAW augments ControlNet-based diffusion with feature-dependent spatial weighting. We first review mask-conditioned diffusion for completeness and then detail the adaptive weighting module used in the reported LAW objective. See Fig. 4 for the final pipeline.

#### 3.1.1 Preliminaries: Mask-Conditioned Diffusion

LAW builds on latent diffusion models (Rombach et al., 2022) that denoise in a compressed latent space. Given an image $\mathbf{x}_0$, a pretrained encoder $\mathcal{E}$ maps it to latent $\mathbf{z}_0 = \mathcal{E}(\mathbf{x}_0)$. The forward diffusion process adds Gaussian noise over $T$ steps:

$$\mathbf{z}_t = \sqrt{\bar{\alpha}_t}\mathbf{z}_0 + \sqrt{1 - \bar{\alpha}_t}\boldsymbol{\epsilon}, \quad \boldsymbol{\epsilon} \sim \mathcal{N}(\mathbf{0}, \mathbf{I}), \tag{1}$$

where $\bar{\alpha}_t$ controls the noise schedule. A denoising network $\epsilon_\theta$ predicts the noise given $\mathbf{z}_t$, timestep $t$, and conditioning. For mask-conditioned synthesis, ControlNet (Zhang et al., 2023) based models inject spatial control via a segmentation mask $\mathbf{m} \in [0, 1]^{H \times W}$ by adding trainable copies of the encoder with zero-initialized connections. The standard training objective minimizes:

$$\mathcal{L}_{\text{base}} = \mathbb{E}_{t, \mathbf{z}_0, \boldsymbol{\epsilon}} \left[ \|\epsilon_\theta(\mathbf{z}_t, t, \mathbf{m}) - \boldsymbol{\epsilon}\|_2^2 \right]. \tag{2}$$

In medical images, lesions occupy small regions (often <10% of pixels) against vast backgrounds. Uniform loss weighting causes models to prioritize background reconstruction, leading to poor lesion-mask alignment. Prior work (Du et al., 2023; Qiu et al., 2025b) addresses this via ratio-based spatial weighting:

$$w_{\text{ratio}}(\mathbf{m}) = \mathbf{m} \cdot (1 - r) + \bar{\mathbf{m}} \cdot r, \tag{3}$$

where $\bar{\mathbf{m}} = 1 - \mathbf{m}$ and $r = \frac{|\mathbf{m}|}{HW}$ is the lesion-area ratio. This linear prior balances lesion and background contributions based on their relative coverage, but remains oblivious to local feature complexity or denoising difficulty.

### 3.1.2 Adaptive Weight Learning

LAW learns spatially adaptive weights from the denoiser prediction and the conditioning segmentation mask. Let $\mathbf{f}_t = \hat{\boldsymbol{\epsilon}}_t^S \in \mathbb{R}^{B \times C \times h \times w}$ denote the denoiser prediction at the latent resolution. The conditioning mask is resized to the same spatial resolution with nearest-neighbor interpolation and reduced to one channel when needed, yielding $\mathbf{m} \in [0,1]^{B \times 1 \times h \times w}$. LAW predicts a delta map $\boldsymbol{\delta} \in [0,1]^{B \times 1 \times h \times w}$ that modulates the ratio-based prior:

$$\boldsymbol{\delta} = \sigma\left(\frac{1}{\tau} \cdot \phi(\mathbf{f}_t, \mathbf{m})\right), \tag{4}$$

where $\phi$ is a lightweight convolutional network (3 layers, 32 hidden channels), $\sigma$ denotes the sigmoid function, and $\tau = 3.0$ is a temperature parameter. The delta map adjusts the base weights via:

$$\boldsymbol{\mu} = 1.0 + \gamma \cdot (2\boldsymbol{\delta} - 1.0), \tag{5}$$

where $\gamma = 0.2$ in our implementation, producing multipliers confined to $[0.8, 1.2]$ after clamping. The final adaptive weights combine the ratio prior with learned adjustments:

$$w_{\text{adapt}} = w_{\text{ratio}}(\mathbf{m}) \odot \boldsymbol{\mu}, \tag{6}$$

where $\odot$ denotes element-wise multiplication. To ensure stability and prevent model collapse, LAW normalizes to preserve mean magnitude and applies both minimum and maximum weight bounds:

$$w_{\text{final}} = \text{clamp}\left(\frac{w_{\text{adapt}}}{\text{mean}(w_{\text{adapt}})}, w_{\text{min}}, w_{\text{max}}\right), \tag{7}$$

with $w_{\text{min}} = 10^{-3}$ and $w_{\text{max}} = 2.0$. The maximum bound prevents extreme weights from over-emphasizing lesion regions at the expense of background coherence; removing it inflates FID from 52.28 to 80.22 (Table 6) and drops mask-recovery Dice from 0.82 to 0.79 on the same Polyps protocol. The spatially weighted denoising loss becomes:

$$\mathcal{L}_S = \mathbb{E}_{t, \mathbf{z}_0, \boldsymbol{\epsilon}}\left[w_{\text{final}} \odot \|\hat{\boldsymbol{\epsilon}}_t^S - \boldsymbol{\epsilon}\|_2^2\right]. \tag{8}$$

Here the unreduced denoising error map $\|\hat{\boldsymbol{\epsilon}}_t^S - \boldsymbol{\epsilon}\|_2^2$ is kept in $\mathbb{R}^{B \times C \times h \times w}$, so $w_{\text{final}} \in \mathbb{R}^{B \times 1 \times h \times w}$ is broadcast across channels before averaging over channels and spatial locations. To prevent degenerate solutions where $\boldsymbol{\delta}$ ignores the mask, LAW adds a Dice-style regularizer that encourages spatial alignment:

$$\mathcal{L}_{\text{dice}} = 1 - \frac{2 \sum_{i,j} \delta_{ij} m_{ij} + \epsilon_s}{\sum_{i,j} \delta_{ij} + \sum_{i,j} m_{ij} + \epsilon_s}, \tag{9}$$

where $\epsilon_s = 10^{-6}$ prevents division by zero. The reported training objective is therefore

$$\mathcal{L}_{\text{total}} = \mathcal{L}_S + \lambda_{\text{dice}} \mathcal{L}_{\text{dice}}, \tag{10}$$

where $\lambda_{\text{dice}} = 1.0$.

**Inference cost.** LAW is a training-time loss modification. It changes the parameters learned by the denoiser, but not the sampling rate used at test time. During generation we run the trained ControlNet checkpoint with the same DDIM sampler as the uniform-loss baseline, and the delta-map network $\phi$, the ratio weights, and the Dice regularizer are not evaluated inside the denoising loop. The checkpoint may still store $\phi$ so that attention maps can be logged for analysis, but the generated samples in Section 4 do not invoke it. LAW therefore changes neither the number of sampling steps nor the inference-time FLOPs, parameters, or activation memory of the denoiser.

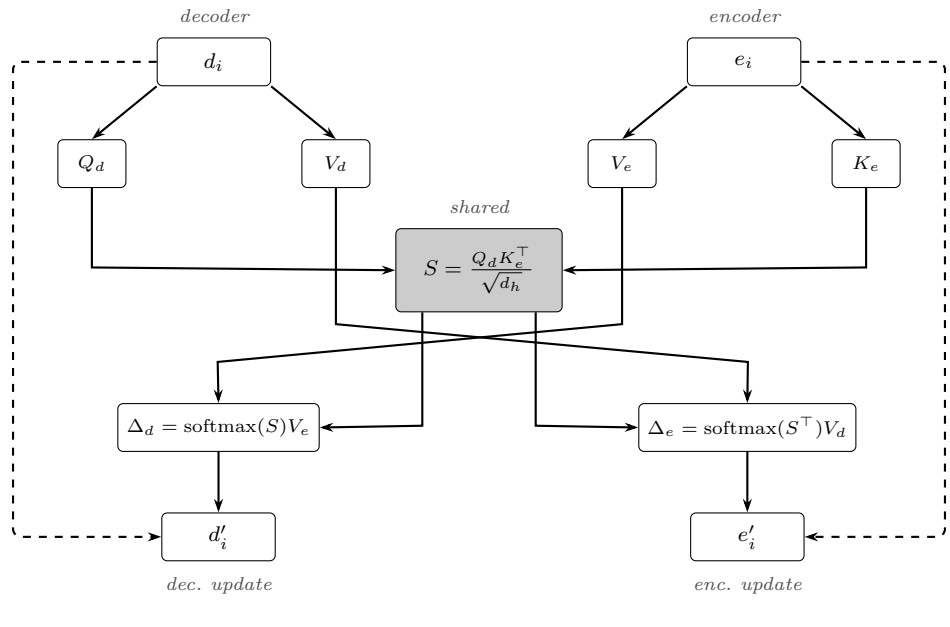

Figure 5: Bidirectional skip attention in ORDER. A single similarity matrix $S$ is computed from decoder queries and encoder keys, then reused for both dec→enc and enc→dec attention via $S$ and $S^\top$, halving the attention cost compared to separate computations.

## 3.2 ORDER: Adaptive Efficient Segmentation

ORDER extends lightweight MK-UNet baselines with selective bidirectional attention and confidence-gated fusion. We summarize the backbone before detailing how the bidirectional skip attention operates (further illustrated in Fig. 5).

### 3.2.1 Preliminaries: Multi-Kernel Lightweight Architecture

ORDER builds on the ultra-lightweight MK-UNet architecture (Rahman & Marculescu, 2025), which achieves 27K parameters through multi-kernel inverted residual (MKIR) blocks. The encoder consists of five stages with multi-kernel depthwise convolutions that capture features at multiple scales. Each stage $i$ produces feature maps $\mathbf{e}_i \in \mathbb{R}^{C_i \times H_i \times W_i}$ with progressively reduced spatial resolution and increased channels. Following (Rahman & Marculescu, 2025), each encoder stage uses MKIR blocks that expand channels, apply parallel depthwise convolutions with multiple kernel sizes $\mathcal{K} = \{1, 3, 5\}$, and project back:

$$\text{MKIR}(\mathbf{x}) = \mathbf{x} + \text{Proj}\left(\sum_{k \in \mathcal{K}} \text{DWConv}_k(\text{Expand}(\mathbf{x}))\right), \tag{11}$$

where Expand uses $1 \times 1$ convolution with expansion factor 2, $\text{DWConv}_k$ applies depthwise convolution with kernel $k$, and Proj projects back to the original channels. This design achieves efficiency through depthwise separable convolutions while capturing multi-scale context. The decoder mirrors the encoder structure with simple additive skip connections.

While MK-UNet achieves high parameter efficiency, it struggles on spatially ambiguous regions where lesion boundaries are unclear. We posit this limitation stems from uniform skip connections that treat all spatial regions equally, failing to modulate fusion strength between easy and hard regions.

### 3.2.2 Selective Bidirectional Skip Attention

ORDER adds bidirectional cross attention to the final decoder stages where semantic information is most critical. Inspired by Reformer (Kitaev et al., 2020), the module computes one similarity matrix that is reused in both directions, avoiding the quadratic cost of separate attentions. Let $\mathbf{d}_i \in \mathbb{R}^{B \times C_i \times H_i \times W_i}$ denote decoder features and $\mathbf{e}_i \in \mathbb{R}^{B \times C_i \times H_i \times W_i}$ the corresponding encoder skip. We flatten both feature maps into $N_i = H_i W_i$ tokens, apply RMSNorm token-wise, reshape back to feature maps, and use $1 \times 1$ convolutions to obtain multi-head projections. The resulting tensors are

$$\mathbf{Q}_d, \mathbf{V}_d \in \mathbb{R}^{B \times H_a \times N_i \times d_h}, \qquad\qquad \mathbf{K}_e, \mathbf{V}_e \in \mathbb{R}^{B \times H_a \times N_i \times d_h}, \qquad (12)$$

where $H_a$ is the number of attention heads and $d_h$ is the per-head width. We first form a shared similarity matrix using the decoder queries and encoder keys:

$$\mathbf{S} = \frac{\mathbf{Q}_d \mathbf{K}_e^\top}{\sqrt{d_h}} \in \mathbb{R}^{B \times H_a \times N_i \times N_i}, \qquad (13)$$

where the softmax is applied over the last dimension. Decoder-side updates attend to encoder values, while encoder-side updates re-use the transposed weights to attend back to decoder values:

$$\mathbf{R}_d = \mathrm{softmax}(\mathbf{S})\mathbf{V}_e,$$
$$\mathbf{R}_e = \mathrm{softmax}(\mathbf{S}^\top)\mathbf{V}_d, \qquad (14)$$

where $\mathbf{R}_d, \mathbf{R}_e \in \mathbb{R}^{B \times H_a \times N_i \times d_h}$ remain in token space. They are then reshaped back to feature maps and projected by $1 \times 1$ convolutions:

$$\Delta_d = \mathrm{Proj}_d(\mathrm{reshape}(\mathbf{R}_d)) \in \mathbb{R}^{B \times C_i \times H_i \times W_i},$$
$$\Delta_e = \mathrm{Proj}_e(\mathrm{reshape}(\mathbf{R}_e)) \in \mathbb{R}^{B \times C_i \times H_i \times W_i}. \qquad (15)$$

Both feature streams are then enhanced residually:

$$\mathbf{d}_i' = \mathbf{d}_i + c_i \Delta_d,$$
$$\mathbf{e}_i' = \mathbf{e}_i + c_i \Delta_e. \qquad (16)$$

**Confidence gate.** Each active skip stage has its own gate module—not a spatial attention map, not a network-wide scalar—that outputs one per-sample scalar modulating both residual updates. We do not interpret $c_i$ as a direct estimator of boundary uncertainty; it is a confidence score for whether the late skip should receive bidirectional refinement. Concretely, the encoder and decoder features are globally average pooled, concatenated channel-wise, and passed through a two-layer $1 \times 1$ MLP:

$$\bar{\mathbf{e}}_i = \mathrm{GAP}(\mathbf{e}_i), \qquad \bar{\mathbf{d}}_i = \mathrm{GAP}(\mathbf{d}_i),$$
$$c_i = \sigma\big(W_2 \, \mathrm{ReLU}\big(W_1[\bar{\mathbf{e}}_i; \bar{\mathbf{d}}_i]\big)\big), \qquad (17)$$

where $[\cdot; \cdot]$ denotes channel concatenation, $\bar{\mathbf{e}}_i, \bar{\mathbf{d}}_i \in \mathbb{R}^{B \times C_i \times 1 \times 1}$, $c_i \in \mathbb{R}^{B \times 1 \times 1 \times 1}$, $W_1$ is a $1 \times 1$ convolution from $2C_i$ channels to $C_i/4$, and $W_2$ is a $1 \times 1$ convolution from $C_i/4$ channels to 1. In the reported ORDER variants, the two active confidence gates add only a few hundred parameters in total. Large $c_i$ values allow the bidirectional residuals to contribute strongly when pooled context indicates that the skip is informative; small $c_i$ values suppress both updates and let the stage fall back toward the original additive skip. The decoder then uses the fused skip $\mathbf{d}_i' + \mathbf{e}_i'$. For other decoder stages, simple additive skip connections suffice. The final segmentation output is produced via $1 \times 1$ convolution followed by sigmoid activation. Training optimizes binary cross-entropy combined with Dice loss:

$$\mathcal{L}_{\mathrm{seg}} = \mathcal{L}_{\mathrm{BCE}}(\hat{\mathbf{y}}, \mathbf{y}) + \mathcal{L}_{\mathrm{Dice}}(\hat{\mathbf{y}}, \mathbf{y}), \qquad (18)$$

where $\hat{\mathbf{y}}$ denotes the predicted mask and $\mathbf{y}$ denotes the ground truth. This combination handles class imbalance while encouraging spatial overlap.

# 4 Experiments

Experiments are organized by method. LAW is evaluated on both mask-conditioned synthesis and downstream augmentation; ORDER is evaluated on efficient segmentation. Unless otherwise stated, all reported overlap metrics use the standard binary definitions

$$\text{Dice}(\hat{y}, y) = \frac{2|\hat{y} \cap y|}{|\hat{y}| + |y|}, \qquad \text{IoU}(\hat{y}, y) = \frac{|\hat{y} \cap y|}{|\hat{y} \cup y|}.$$

## 4.1 LAW

### 4.1.1 Datasets

- **Polyps.** LAW training uses the same 1450 paired image-mask samples as ArSDM and Adaptive Distillation, formed from the standard Kvasir and CVC-ClinicDB diffusion training split (Ali et al., 2023; Qiu et al., 2025b). For evaluation and downstream augmentation, we generate one synthetic image per original training mask, yielding 1450 generated pairs.

- **KiTS19.** The preprocessing pipeline converts KiTS19 (Heller et al., 2021) volumes into 2D axial slices, yielding 5003 train/validation slices and 531 held-out test slices.

- **BRISC.** The BRISC dataset (Fateh et al., 2026) provides 6000 contrast-enhanced T1 MRI slices with expert tumor masks. Our preprocessing produces 5000 train/validation slices and 1000 held-out test slices; after filtering for valid paired masks, 3933 slices remain in the BRISC LAW training pool.

### 4.1.2 Implementation Details

LAW starts from the Stable Diffusion v1.5 ControlNet checkpoint. Images and masks are binarized at threshold 127 and resized to $384 \times 384$; the text prompt is dropped with probability 0.05. On Polyps we train for 1000 steps with batch size 20 and learning rate $10^{-5}$; on KiTS19 and BRISC we use the same optimizer settings and train for 2000 steps. The adaptive module uses 32 hidden channels with $\tau = 3.0$, $\gamma = 0.2$, $\lambda_{\text{dice}} = 1.0$, $w_{\min} = 10^{-3}$, and $w_{\max} = 2.0$. For evaluation, we generate one synthetic image per original training mask, matching the size of the diffusion training set: 1450 pairs on Polyps, 5003 on KiTS19, and 3933 on BRISC. Sampling uses DDIM with 50 steps, $\eta = 0$, and classifier-free guidance scale 10.0. For KiTS19 and BRISC, checkpoint selection is performed on a fixed validation subset carved from the train/validation pool before any held-out test evaluation; in the reported runs this selects step 1000 for KiTS19 and step 900 for BRISC.

### 4.1.3 Baselines and Metrics

- **Baselines.** We compare against uniform-loss ControlNet (Zhang et al., 2023), ArSDM with ratio-based weighting (Du et al., 2023), and adaptive distillation (Qiu et al., 2025b).

- **Image quality.** FID is computed with `pytorch-fid`[1] after resizing both real and generated images to $299 \times 299$ before Inception feature extraction. CLIP-I is the mean cosine similarity between generated-image embeddings and the corresponding real reference set. We report two reference protocols. First, for Polyps and KiTS19, we reproduce the training-distribution-matched setup used by ArSDM and Adaptive Distillation: generated images are compared against the real diffusion training distribution with matched sample counts. Second, for all three datasets, we report held-out test-set FID/CLIP-I by comparing generations from held-out masks against the corresponding held-out real image split. Due to time constraints, we did not recompute the BRISC training-distribution-matched FID/CLIP-I after standardizing the held-out evaluation.

- **Direct spatial control.** Mask-recovery Dice and IoU are measured by running a frozen dataset-specific nnUNet segmenter on every generated image and comparing the predicted mask with the conditioning mask that produced it.

---

[1] `https://github.com/mseitzer/pytorch-fid`

Table 1: LAW image quality under the training-distribution-matched protocol used by prior work. Generated images are compared against the real diffusion training distribution with matched sample counts. We report Polyps and KiTS19 only; due to time constraints we did not recompute the analogous BRISC numbers after standardizing the held-out protocol. All values are mean ± std over five seeded runs.

| Method | Polyps FID↓ | Polyps CLIP-I↑ | KiTS19 FID↓ | KiTS19 CLIP-I↑ |
|---|---|---|---|---|
| ControlNet (Uniform) | 65.54±0.54 | 0.884±0.021 | 69.24±0.65 | 0.833±0.012 |
| ArSDM (Ratio-based) | 98.38±0.61 | 0.843±0.003 | 104.12±0.53 | 0.852±0.012 |
| Adaptive Distillation | 66.40±0.72 | **0.903±0.011** | 70.78±0.57 | 0.839±0.022 |
| LAW (ours) | **52.32±0.70** | 0.895±0.001 | **50.13±0.44** | **0.937±0.001** |

Table 2: LAW image quality on held-out test sets. Generated images are produced from held-out masks and compared against the corresponding held-out real image split.

All values are mean ± std over five seeded runs.

| Method | Polyps FID↓ | Polyps CLIP-I↑ | KiTS19 FID↓ | KiTS19 CLIP-I↑ | BRISC FID↓ | BRISC CLIP-I↑ |
|---|---|---|---|---|---|---|
| ControlNet (Uniform) | 158.13±0.15 | 0.875±0.021 | 144.13±0.31 | 0.837±0.003 | 139.22±0.38 | 0.881±0.025 |
| ArSDM (Ratio-based) | 136.56±0.63 | 0.869±0.091 | 123.24±0.64 | 0.841±0.053 | 123.22±0.04 | 0.891±0.093 |
| Adaptive Distillation | 129.11±0.89 | 0.863±0.076 | 118.76±0.81 | 0.831±0.011 | 131.22±0.74 | 0.886±0.069 |
| LAW (ours) | **108.43±0.71** | **0.887±0.090** | **89.51±0.96** | **0.934±0.051** | **112.58±0.68** | **0.919±0.079** |

Table 3: LAW direct spatial-control results measured by mask recovery. Baselines are uniform-loss Control-Net, ArSDM, and Adaptive Distillation.

| Method | Polyps Mask Dice↑ | Polyps Mask IoU↑ | KiTS19 Mask Dice↑ | KiTS19 Mask IoU↑ | BRISC Mask Dice↑ | BRISC Mask IoU↑ |
|---|---|---|---|---|---|---|
| ControlNet (Uniform) | 0.681±0.013 | 0.582±0.022 | 0.488±0.011 | 0.411±0.023 | 0.511±0.012 | 0.460±0.019 |
| ArSDM (Ratio-based) | 0.783±0.041 | 0.682±0.023 | 0.532±0.030 | 0.453±0.026 | 0.553±0.014 | 0.490±0.022 |
| Adaptive Distillation | 0.794±0.011 | 0.701±0.014 | 0.558±0.015 | 0.478±0.019 | 0.596±0.011 | 0.520±0.026 |
| LAW (ours) | **0.825±0.003** | **0.733±0.005** | **0.586±0.009** | **0.502±0.011** | **0.629±0.014** | **0.547±0.013** |

- **Downstream augmentation.** For each diffusion baseline, we generate one synthetic image per original training mask, yielding matched synthetic pools of 1450 pairs on Polyps, 5003 on KiTS19, and 3933 on BRISC. Table 4 compares the same nnUNet segmenter trained on real data only or on the real pool augmented with each method's full synthetic pool. Table 5 reports the Polyps LAW dose-response obtained by subsampling 250, 500, 1000, or 1450 pairs from the full LAW pool. Validation remains real-only, and the diffusion model itself is trained only on the real training split and does not use held-out real images during training. We report five-seed mean±std mDice.

### 4.1.4 Results

**Image quality across datasets.** Table 1 first confirms that LAW improves the training-distribution-matched protocol used by prior work, lowering Polyps FID from 65.54±0.54 to 52.32±0.70 and KiTS19 FID from 69.24±0.65 to 50.13±0.44. The more practically relevant held-out protocol in Table 2 preserves the same ranking and shows substantial generalization gains: LAW lowers Polyps FID from 158.13±0.15 to 108.43±0.71, KiTS19 FID from 144.13±0.31 to 89.51±0.96, and BRISC FID from 139.22±0.38 to 112.58±0.68. LAW also attains the highest held-out CLIP-I on all three datasets.

**Direct spatial control.** Table 3 reports mask-recovery Dice and IoU, which are the more task-aligned control metrics. On Polyps, LAW raises mask Dice from 0.681±0.013 for uniform ControlNet to 0.825±0.003. On KiTS19, it improves mask Dice from 0.488±0.011 to 0.586±0.009, and on BRISC it improves mask Dice from 0.511±0.012 to 0.629±0.014. Figures 6 and 7 show representative Polyps, KiTS19, and BRISC generations.

**Downstream augmentation.** Table 4 compares the same real training pool augmented by matched synthetic pools from each diffusion method. LAW gives the best downstream mDice on Polyps and BRISC, raising Polyps from 71.7±0.4 to 74.1±0.8 and BRISC from 87.7±0.2 to 89.6±0.4. On KiTS19, LAW reaches 73.2±0.6, which is within 0.1 mDice of adaptive distillation's best score of 73.3±0.5. These are moderate

Table 4: Downstream segmentation mDice when training the same nnUNet segmenter on real data only or on the same real pool augmented with one full synthetic pool per method. Each synthetic pool contains one generated image per original training mask: 1450 pairs on Polyps, 5003 on KiTS19, and 3933 on BRISC.

| All rows report mean ± std over five seeds. | | | |
|---|---|---|---|
| Training Data | Polyps mDice↑ | KiTS19 mDice↑ | BRISC mDice↑ |
| Real only | 71.7±0.4 | 70.4±2.9 | 87.7±0.2 |
| Real + ControlNet | 72.3±1.2 | 71.1±0.8 | 88.1±0.3 |
| Real + ArSDM | 73.5±1.1 | 72.3±0.5 | 88.7±0.3 |
| Real + Adaptive Distillation | 72.6±0.9 | **73.3±0.5** | 89.0±0.4 |
| Real + LAW | **74.1±0.8** | 73.2±0.6 | **89.6±0.4** |

Table 5: Polyps downstream dose-response under LAW-based synthetic augmentation. Rows vary the number of LAW-generated image-mask pairs subsampled from the full 1450-pair Polyps synthetic pool and mixed into the nnUNet training pool.

| Training control | Synthetic images | Mean Dice↑ |
|---|---|---|
| Real only | 0 | 71.7±0.4 |
| Real + LAW | 250 | 72.3±0.6 |
| Real + LAW | 500 | 72.5±0.6 |
| Real + LAW | 1000 | 73.8±0.7 |
| Real + LAW | 1450 | **74.1±0.8** |

rather than dramatic downstream gains, which is why we present them as dataset-dependent augmentation benefits instead of a universal improvement claim.

**Dose response.** Table 5 shows a monotonic Polyps dose-response as the number of LAW pairs increases. Moving from 250 to 500 pairs gives a small gain, from 72.3±0.6 to 72.5±0.6, while larger pools reach 73.8±0.7 at 1000 pairs and 74.1±0.8 at the full 1450-pair pool. Since the 250/500/1000/1450 rows are all subsampled from the same full LAW pool, the trend suggests that broader synthetic coverage helps until the gain begins to saturate.

### 4.1.5 Ablation Studies

Table 6: Component ablation of LAW on Polyps synthesis. We report single-run image-quality metrics together with qualitative training stability. The last row isolates the max-weight clamp by disabling it on the full LAW configuration.

| Configuration | FID↓ | CLIP-I↑ | Stability |
|---|---|---|---|
| Adaptive Distillation (Qiu et al., 2025b) | 66.60 | **0.901** | Stable |
| + Delta map (no norm) | 70.34 | 0.854 | Unstable |
| + Normalization | 70.11 | 0.876 | Stable |
| + Min-weight clamp | 66.51 | 0.881 | Stable |
| + Dice regularizer (LAW) | **52.28** | 0.893 | Stable |
| LAW w/o max-weight clamp | 80.22 | 0.893 | Stable |

**LAW components.** Table 6 shows that the gain does not come from learned weighting alone. The raw delta-map variant shifts emphasis too aggressively and destabilizes optimization because the average loss scale is no longer controlled. Mean-preserving normalization fixes that scale mismatch, while the minimum clamp prevents near-zero weights from effectively silencing easy background regions. The largest quality jump appears only after adding the Dice regularizer, which encourages the learned weighting map to stay aligned with the intended lesion support instead of drifting toward arbitrary high-error regions. Finally, removing only the upper clamp leaves training numerically stable but sharply worsens FID, indicating that uncontrolled lesion upweighting harms global anatomical coherence even when optimization does not diverge. Figure 8 visualizes how the learned LAW maps sharpen over training.

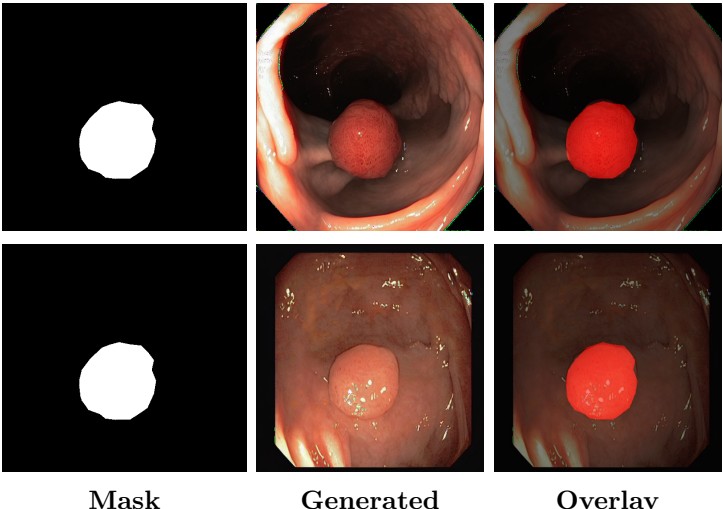

**Mask**   **Generated**   **Overlay**

Figure 6: Qualitative Polyps comparison. Top: Adaptive Distillation baseline. Bottom: LAW. LAW better preserves lesion placement while keeping the background anatomy coherent.

## 4.2 ORDER

### 4.2.1 Datasets

- **Polyps.** ORDER uses a cleaned 2026-image training pool formed from the 1450 Polyps training pairs together with all ETIS-LaribPolypDB (196 pairs) and CVC-ColonDB (380 pairs) cases. For each seed $s \in \{0, \ldots, 4\}$ we hold out 10% of this pool for validation, giving 1823 train and 203 validation images. External testing uses only the disjoint Kvasir (100 images) and CVC-ClinicDB (62 images) subsets.

- **BRISC.** We also train ORDER on BRISC using the same five seeds with a 90/10 train/validation split over the 5000-image training pool and evaluate on the 1000-image BRISC test set.

### 4.2.2 Implementation Details

ORDER uses an ultra-compact backbone with channels $[4, 8, 16, 24, 32]$, two attention heads, head width 32, and bidirectional attention on decoder skip stages $\{0, 1\}$ in the reported main configuration. Table 10 shows that $\{1, 2\}$ attains the best raw Dice on an internal Polyps development split, but Table 9 motivated our final choice of $\{0, 1\}$ as the better practical trade-off between accuracy, latency, and peak memory. Training uses $256 \times 256$ inputs, batch size 16, 100 epochs, BCE+Dice loss, AdEMAMix optimizer with learning rate $10^{-4}$ and weight decay $10^{-4}$, cosine annealing, and random horizontal flips, vertical flips, and rotations.

**Reproducibility.** All Polyps segmentation rows in Tables 7 and 8 are five-seed runs under the same wrapper, train/validation split rule, augmentation pipeline, and evaluation code. Mean Dice/IoU are obtained by averaging the per-subset metrics for Kvasir and CVC-ClinicDB. BRISC uses the same optimizer, augmentation, and split recipe as the Polyps study, with evaluation on the held-out BRISC test set.

### 4.2.3 Baselines and Metrics

- **Baselines.** We compare ORDER against MK-UNet (Rahman & Marculescu, 2025), EGE-UNet (Ruan et al., 2023), PraNet (Fan et al., 2020), and a heavy nnUNet 2D reference. We also add two matched compact controls on the same ORDER backbone without bidirectional attention: ORDER backbone + SE skip, which uses squeeze-excitation recalibration (Hu et al., 2018), and ORDER backbone + CBAM skip, which uses sequential channel/spatial attention (Woo et al., 2018). Both controls are inserted on the same skip stages, $\{0, 1\}$, and trained under the same five-seed wrapper as the reported ORDER$\{0, 1\}$ model.

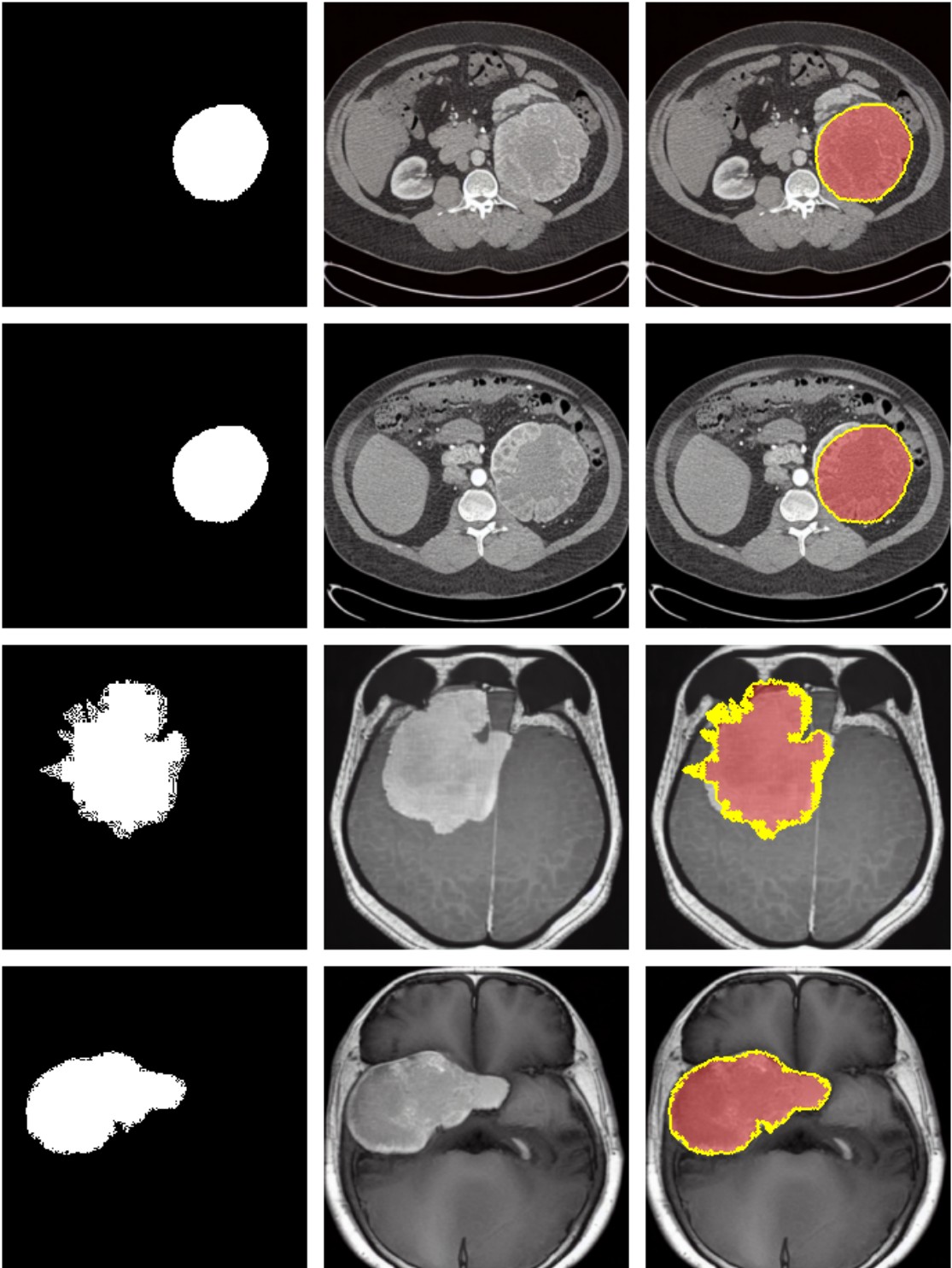

Figure 7: Additional LAW generations from KiTS19 and BRISC. From top to bottom, the four rows show two KiTS19 examples followed by two BRISC examples; columns are conditioning mask, generated image, and overlay.

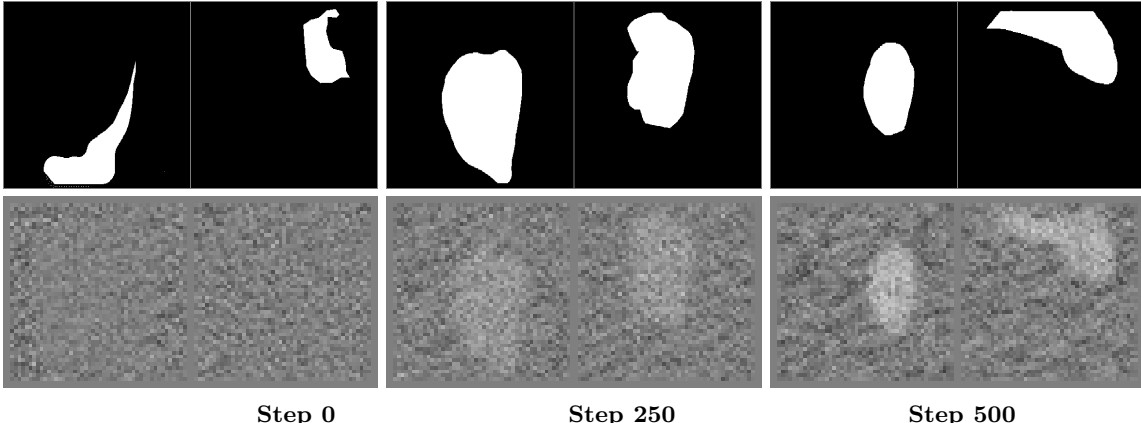

**Step 0**  **Step 250**  **Step 500**

Figure 8: Visualization of learned LAW delta maps during training. Top: conditioning masks. Bottom: learned attention maps. The module starts nearly uniform and progressively concentrates on lesion regions and boundaries.

Table 7: Segmentation comparison on the cleaned Polyps protocol and BRISC. Values are mean ± std over five seeded runs.

| Method | Params | FLOPs | Polyps Dice↑ | Polyps IoU↑ | BRISC Dice↑ | BRISC IoU↑ |
|---|---|---|---|---|---|---|
| nnUNet | 31.0M | 109.2G | 87.2±0.4 | 81.2±0.4 | 87.7±0.2 | 80.5±0.3 |
| PraNet | 24.2M | 11.3G | 90.1±0.7 | 84.8±0.7 | 84.1±0.1 | 76.8±0.3 |
| EGE-UNet | 0.51M | 0.62G | 65.5±0.3 | 53.6±0.8 | **78.3±0.6** | **70.1±0.9** |
| MK-UNet | 27K | 0.10G | 70.3±1.5 | 59.9±1.7 | 75.4±1.2 | 66.4±1.4 |
| ORDER$\{0,1\}$ (ours) | 42K | 0.11G | **76.3±1.9** | **67.2±2.0** | 77.4±0.8 | 68.1±0.7 |

- **Metrics.** We report per-subset and mean mDice/mIoU on thresholded binary masks, together with parameter counts and FLOPs estimated at $256^2$ resolution by the same training wrappers. For the runtime study, we additionally report latency, throughput, and peak memory from a fixed local profiling setup; these measurements are hardware-dependent and should be interpreted comparatively rather than as universal deployment numbers.

#### 4.2.4 Results

**Standard comparison.** Table 7 separates heavyweight references from the lighter models below the divider. Within this compact comparison set, ORDER$\{0,1\}$ gives the best Polyps Dice and IoU while remaining close to MK-UNet in scale: 42K vs. 27K parameters and 0.11 vs. 0.10 GFLOPs. Relative to the matched MK-UNet baseline, it raises Polyps mDice from 70.3±1.5 to 76.3±1.9 and mIoU from 59.9±1.7 to 67.2±2.0, while BRISC rises from 75.4±1.2 to 77.4±0.8 and from 66.4±1.4 to 68.1±0.7. Figure 9 shows representative qualitative outputs on both datasets. nnUNet and PraNet remain stronger in absolute accuracy at far larger budgets. EGE-UNet leads ORDER on BRISC by 0.9 mDice (78.3 vs. 77.4) despite using 12× more parameters (0.51M vs. 42K); on Polyps the ordering is reversed, with ORDER outperforming EGE-UNet by 10.8 mDice.

**Attention controls.** Table 8 compares the reported ORDER$\{0,1\}$ module against standard attention mechanisms on the same compact backbone. The SE skip control recovers most of the gain over the plain backbone, which confirms that simple recalibration already helps in this regime. However, ORDER$\{0,1\}$ still gives the best mean Dice and IoU. CBAM trails both SE and ORDER. This isolates the contribution of selective bidirectional skip interaction: the gain is not just from adding any generic attention block, but from the specific decoder–encoder exchange used by ORDER. We therefore make the narrower empirical claim that ORDER helps by selectively refining late skips, rather than claiming that the scalar confidence gate directly proves boundary uncertainty localization.

Table 8: Attention control study on the same ORDER backbone without bidirectional attention. SE and CBAM are inserted on the same skip stages, $\{0, 1\}$, as the reported ORDER$\{0, 1\}$ model. All values are mean $\pm$ std over five seeded runs.

| Method | Params | FLOPs | Kv. Dice↑ | Kv. IoU↑ | Clin. Dice↑ | Clin. IoU↑ | Mean Dice↑ | Mean IoU↑ |
|---|---|---|---|---|---|---|---|---|
| ORDER backbone | 26K | 0.10G | 72.4±2.2 | 61.9±2.2 | 68.1±0.9 | 58.0±1.3 | 70.3±1.5 | 59.9±1.7 |
| ORDER backbone + SE skip | 26K | 0.12G | 77.3±2.9 | 67.7±3.1 | 73.3±2.5 | 64.1±2.9 | 75.3±2.7 | 65.9±2.9 |
| ORDER backbone + CBAM skip | 27K | 0.12G | 75.8±1.8 | 65.8±2.0 | 70.9±1.3 | 61.3±1.7 | 73.3±1.6 | 63.5±1.8 |
| ORDER$\{0, 1\}$ (ours) | 42K | 0.11G | **79.1±1.6** | **69.6±1.7** | **73.6±2.4** | **64.8±2.4** | **76.3±1.9** | **67.2±2.0** |

Table 9: Runtime and memory profiling at $256^2$ resolution on one fixed local setup. These numbers are hardware-dependent and are intended as relative comparisons across models rather than universal deployment claims.

| Model | Params | FLOPs | ms/img↓ | img/s↑ | Peak MB↓ |
|---|---|---|---|---|---|
| MK-UNet | 26,307 | 0.095G | 0.230 | 4356.4 | 180.2 |
| ORDER$\{0, 1\}$ | 42,265 | 0.112G | 0.399 | 2509.0 | 559.5 |
| ORDER$\{1, 2\}$ | 35,793 | 0.132G | 2.954 | 338.5 | 8290.8 |
| ORDER backbone + SE | 26,467 | 0.095G | 0.238 | 4197.2 | 189.2 |
| ORDER backbone + CBAM | 26,663 | 0.096G | 0.254 | 3940.1 | 189.4 |
| nnUNet | 31,037,633 | 109.220G | 2.667 | 375.0 | 1901.7 |

**Runtime and memory.** Table 9 explains why the main paper reports ORDER$\{0, 1\}$ rather than ORDER$\{1, 2\}$. Although the internal Polyps development split in Table 10 favors $\{1, 2\}$ for raw Dice, the $\{0, 1\}$ variant is much faster on our profiling setup (0.399 vs. 2.954 ms/img) and avoids the large peak-memory cost of ORDER$\{1, 2\}$ (559.5 vs. 8290.8 MB). ORDER$\{0, 1\}$ is therefore the main reported configuration.

### 4.2.5 Ablation Studies

Table 10: Five-seed ablation of which decoder skip indices use bidirectional attention on an internal Polyps development split. Values report mean $\pm$ std. Index 0 is the deepest decoder skip and index 3 is the shallowest. The best raw Dice occurs at $\{1, 2\}$, while the main paper reports $\{0, 1\}$ for the runtime and memory reasons in Table 9. These development split numbers are not directly comparable to Table 7, which evaluates on the disjoint external Kvasir and CVC-ClinicDB test sets.

| Stages with BiAttn | Params | mDice↑ |
|---|---|---|
| None | 26K | 73.4±0.9 |
| Shallow skips {2,3} | 31K | 73.2±0.7 |
| Mixed skips {1,2} | 36K | **75.7±1.1** |
| Deep skips {0,1} | 42K | 72.7±0.8 |
| All skips {0,1,2,3} | 47K | 16.7±1.4 |

**Skip-stage selection.** Table 10 shows that the placement of bidirectional attention changes the accuracy-efficiency trade-off. On this internal Polyps development split, the mixed pair $\{1, 2\}$ gives the best raw Dice, while the deeper pair $\{0, 1\}$ trails it in accuracy but is the variant retained for the main paper because Table 9 shows a much better measured latency and memory profile. The apparent gap between the development split number (72.7 for $\{0, 1\}$) and the main table (76.3) reflects a difference in evaluation protocol: the ablation uses an internal held-out subset of the Polyps training pool, while Table 7 reports on the disjoint external Kvasir and CVC-ClinicDB test sets. The shallow pair $\{2, 3\}$ still sees high-resolution features, but those skips carry weaker semantics and therefore benefit less from cross-stream matching. Applying bidirectional attention to every skip is much worse: early low-level skips inject noisy correlations, and the accumulated cross-stream updates overwhelm the lightweight decoder instead of helping it.

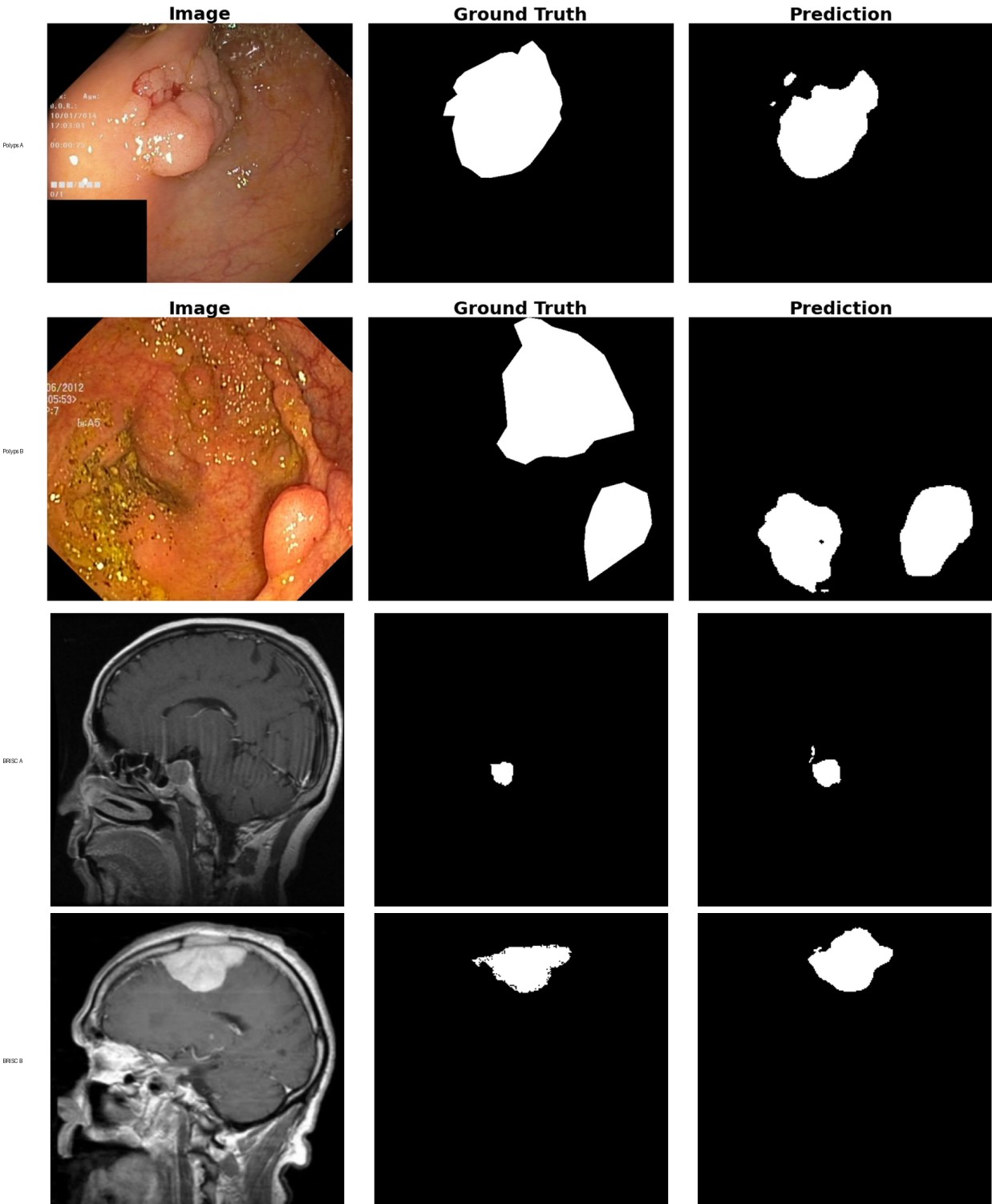

Figure 9: Qualitative ORDER results on Polyps and BRISC. The first two rows are Polyps examples and the last two rows are BRISC examples; each row shows input image, ground-truth mask, and prediction.

## 5    Conclusion

Adaptive spatial weighting is useful here as a design principle, not as a claim of a single unified mechanism. LAW improves both held-out image realism and mask adherence, and its synthetic data gives moderate downstream gains on Polyps and BRISC while remaining competitive on KiTS19. The reported ORDER$\{0, 1\}$ configuration improves the accuracy-efficiency trade-off of an ultra-compact segmenter on the cleaned Polyps benchmark and remains above the matched MK-UNet baseline on BRISC under the same training recipe, while a separate ORDER$\{1, 2\}$ variant attains higher internal development Dice at a much worse runtime and memory cost. The main lesson is practical: learning where to spend capacity is valuable in both diffusion and segmentation when the foreground is sparse. The remaining gaps are also clear from the experiments. ORDER still sits below heavyweight segmenters in absolute accuracy, and broader multimodal validation together with stronger control metrics remain future work.

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

## A   Additional Implementation Details

This appendix collects the implementation details that support reproducibility but would interrupt the main narrative if kept inline. We summarize the fixed hyperparameters used in the reported LAW and ORDER runs, including optimization settings, resolutions, and the active skip configuration retained for the final ORDER comparisons.

Table 11: LAW hyperparameters used in the reported runs.

| Setting | Value |
| --- | --- |
| Backbone | Stable Diffusion v1.5 + ControlNet |
| Training resolution | $384 \times 384$ |
| Prompt drop probability | 0.05 |
| Batch size | 20 |
| Learning rate | $10^{-5}$ |
| Polyps training steps | 1000 |
| KiTS19 / BRISC training steps | 2000 |
| Adaptive hidden channels | 32 |
| Temperature $\tau$ | 3.0 |
| Multiplier scale $\gamma$ | 0.2 |
| Minimum / maximum weight | $10^{-3}$ / 2.0 |
| Dice regularizer weight $\lambda_{\mathrm{dice}}$ | 1.0 |
| Sampler | DDIM |
| Sampling steps | 50 |
| CFG scale | 10.0 |
| Samples per evaluation run | One per conditioning mask |

Table 12: ORDER hyperparameters used in the reported runs.

| Setting | Value |
| --- | --- |
| Backbone | Compact MK-UNet-style (5 stages, MKIR blocks) |
| Channel widths | $[4, 8, 16, 24, 32]$ |
| Active bidirectional skip indices | $\{0, 1\}$ for reported ORDER; $\{1, 2\}$ for dev-ablation best Dice |
| Attention heads / head width | 2 / 32 |
| Input resolution | $256 \times 256$ |
| Loss | BCE + Dice |
| Optimizer | AdEMAMix |
| Learning rate / weight decay | $10^{-4}$ / $10^{-4}$ |
| Scheduler | Cosine annealing to $10^{-6}$ |
| Batch size | 16 |
| Epochs | 100 |
| Augmentations | Random horizontal flip, vertical flip, rotation |
| Polyps train/val split | 1823 / 203 per seed |
| BRISC train/val split | 4500 / 500 per seed |

