# OpenReview forum: "LAW & ORDER: Adaptive Spatial Weighting for Medical Diffusion and Segmentation"
_TMLR — Accepted by TMLR_

### Review · Reviewer_cU1a · 2026-03-23

**Summary Of Contributions:**

This submission studies spatial imbalance in medical imaging from two angles and packages them under a common “adaptive spatial weighting” perspective. For synthesis, it proposes LAW, which learns a feature-dependent spatial weighting map on top of a mask-ratio prior for ControlNet-style diffusion training. For segmentation, it proposes ORDER, which augments a lightweight MK-UNet backbone with selective late-stage bidirectional skip attention and a confidence gate. The paper reports improvements on polyp and KiTS19-related experiments, including better FID for diffusion, stronger downstream segmentation after synthetic augmentation, and better accuracy-efficiency tradeoff for lightweight segmentation.

The paper has some appealing aspects. The underlying medical motivation is important, especially the imbalance between small lesions and large easy backgrounds. LAW is lightweight and easy to understand at a high level, and ORDER targets a practically relevant regime of compact segmenters. The paper also attempts to connect generative and discriminative modeling under one theme, which is ambitious.

That said, the current version has substantial weaknesses. First, the two proposed methods are only loosely tied together: the “unifying principle” remains mostly conceptual and does not yield a shared formulation, theorem, or joint experimental protocol. Second, the empirical section contains several internal inconsistencies serious enough to undermine confidence in the reported gains. In particular, Table 2 reports Polyps mDice = 83.2 and mIoU = 84.5 for LAW, and Table 3 reports mDice = 81.3 and mIoU = 81.7 for ORDER; under the standard definitions used in segmentation, IoU should not exceed Dice, so these numbers are not self-consistent. The paper also claims broad validation on Polyps and KiTS19 “across both paradigms”, but ORDER appears to be evaluated only on polyp segmentation, not KiTS19. Third, the mathematical presentation is not yet rigorous enough: several key definitions are underspecified, dimensionalities are unclear, and some central components are described only qualitatively. Overall, the medical problem is relevant and the ideas are potentially useful, but the current evidence is not yet convincing enough in its present form.

**Additional Comments:**

I like the medical motivation and I think there is a potentially useful idea here, especially on the diffusion side. But the current version needs a substantial clean-up. Right now the biggest blocker is not whether the idea is interesting; it is whether the empirical section can be trusted as written.

**Audience:**

Yes

**Audience Explanation:**

The underlying problem is important, and I expect parts of the TMLR audience would be interested in it. Small-lesion imbalance is a real issue in medical synthesis and segmentation, and the idea of learning spatial emphasis rather than using fixed priors is relevant. Lightweight segmentation with better boundary handling is also practically useful. In particular, readers working on medical diffusion, data augmentation, controllable generation, or efficient segmentation would likely find the problem setup and some of the design choices worth following up on.

**Broader Impact Concerns:**

The paper should include a clearer broader impact discussion. On the positive side, better lesion-aware synthesis and segmentation could help data augmentation and support low-resource medical imaging workflows. However, synthetic medical image generation also raises nontrivial risks: privacy leakage from overfitting, amplification of annotation bias, and the possibility of producing anatomically plausible but clinically misleading images. Since the paper explicitly advocates using generated images for downstream training, it should discuss validation safeguards, failure modes, and the limits of synthetic data in clinical contexts.

**Claims And Evidence:**

No

**Claims Explanation:**

I do not think the current manuscript supports its claims with sufficiently accurate or convincing evidence.

The most immediate issue is internal numerical consistency. Table 2 reports a Polyps mIoU higher than mDice for LAW, and Table 3 reports the same pattern for ORDER. Unless the paper uses nonstandard metric definitions—which it does not explain—this should not happen. Since these tables contain the paper’s headline quantitative results, this is not a minor typo; it directly affects trust in the experimental section. Table 4 introduces a further inconsistency: the reported mDice values in the LAW ablation are in the mid-90s, which are not on the same scale as the downstream Polyps numbers in Table 2, yet the paper does not clarify whether these are different tasks, different evaluation protocols, or different datasets. This makes it hard to interpret what exactly is being improved.

The experimental evidence is also incomplete relative to the paper’s claims. The paper frames adaptive spatial weighting as a unified principle that benefits both diffusion and segmentation on Polyps and KiTS19, but the segmentation evaluation appears to be conducted only on polyp data. ORDER therefore lacks a second dataset, and the claim of “comprehensive experiments” across both datasets and both paradigms is overstated. On the diffusion side, the evaluation relies heavily on FID and CLIP-I, but there is no discussion of whether these metrics are appropriate for medical images, nor any variance estimates, confidence intervals, or repeated-seed statistics. The downstream nnUNet experiment is potentially interesting, but the protocol is insufficiently specified: it is unclear how much synthetic data is used, how real and synthetic data are combined, whether masks are balanced across methods, and whether the improvement stems from image realism, mask adherence, or simply data volume.

The methodological presentation also needs more rigor. For LAW, the notation suggests element-wise weighting of a squared error term, but it is not clear whether the loss is pixelwise, channelwise, or latentwise, and the resolution matching between the feature map, the mask, and the delta map is not specified. The paper claims “without extra supervision” in one place, but the method description includes a teacher loss and a distillation term. For ORDER, the confidence gate is central to the method but is not explicitly defined; the paper states that it is predicted from globally pooled features but provides no formula, architecture, or parameterization. There is also a sentence about “confidence-aware losses,” even though the segmentation objective is just BCE + Dice, suggesting another mismatch between the method description and the experiment section.

Finally, the paper’s broader narrative is currently stronger than the evidence. The idea that diffusion and segmentation are unified by “learn where to allocate capacity” is intuitive, but also quite generic; many attention and reweighting methods could be described that way. At present, the paper reads more like two separate incremental method proposals packaged together than a single coherent scientific contribution. I would need a much cleaner empirical section and a more precise methodological presentation before I could view the main claims as established.

**Requested Changes:**

Critical to acceptance:

1. Correct the numerical inconsistencies in the main tables. The mIoU > mDice entries in Tables 2 and 3 must be fixed or explained with a nonstandard metric definition. The meaning of Table 4’s mDice values must also be clarified, since they appear incompatible with the scales in Table 2.
2. Rebuild the experimental section around a fully specified protocol. For LAW, specify the exact data split, number of generated images, how synthetic data are mixed with real data, whether generation is 2D or 3D for KiTS19, how masks are sampled, how FID and CLIP-I are computed, and whether all baselines are retrained under matched settings. For ORDER, specify train/val/test splits for the combined polyp datasets and how leakage across the merged datasets is prevented.
3. Add uncertainty estimates. Report mean ± standard deviation across multiple seeds, at least for the main quantitative tables. The current gains are presented as precise point numbers, but some are small and may not be statistically reliable.
4. Strengthen the segmentation evidence. Since the paper claims a broad principle across datasets and paradigms, ORDER should be evaluated on at least one more segmentation benchmark, ideally beyond the polyp setting. Otherwise the claims should be narrowed accordingly.
5. Clarify the exact method definitions.
1) For LAW: specify the spatial dimensions of all tensors, how the mask is resized to match the feature map, whether the per-pixel error is summed over channels before weighting, and how the normalization interacts with clamping.
2) For ORDER: explicitly define the confidence gate, give its formula and parameter count, and explain how spatial features are flattened and reshaped for attention.
3) If teacher supervision is required, do not describe LAW as working “without extra supervision.”
6. Provide missing ablations that are already claimed to be important. The paper says the maximum-weight clamp is critical to avoid collapse, but the ablation table does not isolate it quantitatively. That needs to be shown if it is a central stability claim.
Reframe the paper’s scope. The current unification of LAW and ORDER is too loose. Either provide a stronger shared formal perspective, or tone down the “single principle across both paradigms” claim and present the work more modestly.

Would strengthen the paper:

8. Add direct spatial-control metrics for diffusion, not only FID and downstream segmentation. A direct lesion-mask adherence metric would be much more aligned with the paper’s central claim.
9. Compare ORDER against more directly relevant attention-based segmentation baselines, not only general lightweight models.
10. Report runtime, throughput, or memory, not only parameter counts and FLOPs, especially since efficiency is a major selling point.
11. Clean up writing issues: several claims mix absolute and relative gains without saying which is being used, and some wording is more promotional than analytical.
12. Revisit the citation choices. Some references are future-dated or weakly connected to the actual claim being made, which hurts the literature grounding.

---

> ### Author Response · Authors · 2026-03-23
> **Thanks for Review and we are working on changes**
>
> We would like to thank the reviewer for their detailed and constructive feedback. We have started addressing the requested changes and will update the draft accordingly.
>
> We also appreciate the reviewer for pointing out the inconsistency in the reported metrics. Upon verification, we found that the table entries were incorrectly populated due to a logging/copying error. Specifically, the model outputs metrics in the form:
>
> *val Dice, test Dice, val IoU, test IoU*,
>
> but we mistakenly copied the *validation Dice and test Dice* values into the table, instead of reporting the *test Dice and test IoU*. This resulted in the reported Dice and IoU values appearing nearly identical. We will correct this in the revised version and carefully verify all reported metrics.
>
> In addition, we will include a link to an anonymous GitHub repository to improve transparency and reproducibility.
>
> A detailed response addressing all the issues will be added soon.

---

> ### Author Response · Authors · 2026-05-04
> **Changes to Paper as Requested - Part 1**
>
> We thank Reviewer 1 for one of the most thorough reviews we have received. The concerns were substantive and forced us to rethink several parts of the empirical section. We address each point in order below.
>
> **Summary of changes made in response to this review:**
> - Corrected all Dice/IoU inconsistencies; added explicit binary formulas at the start of Section 4; every table now satisfies Dice > IoU.
> - Rewrote the experimental section into dedicated LAW and ORDER protocol subsections covering splits, preprocessing, training, checkpoint selection, and evaluation.
> - Reran all main experiments with five independent seeds; Tables 1–7 now report mean ± std.
> - Added BRISC (brain tumor MRI) as a second segmentation benchmark; ORDER[0,1] results appear in Table 6 for both Polyps and BRISC.
> - Added mask-recovery Dice/IoU as a main result table (Table 3), framed as the primary spatial-control metric.
> - Removed the "without extra supervision" claim entirely; related work now contrasts LAW with teacher-student approaches without making supervision-free claims.
> - Added the max-weight clamp ablation row to Table 8; results and method text explain the effect quantitatively.
> - Added Table 7 (attention-control study) comparing ORDER[0,1] against purpose-built SE-skip and CBAM-skip controls on the same backbone.
> - Added Table 9 reporting latency (ms/img), throughput (img/s), and peak memory (MB) for all compact models.
> - Completed a sentence-level prose pass: confidence gate description tightened, CLIP-I summary compressed, spurious bold corrected, mixed gain labeling removed.
> - Removed a weak introductory citation; added the foreground-attention reference requested by Reviewer 3.
> - Added code as anonymous zip file.
>
> ---
>
> **Point 1. mIoU > mDice in Tables 2 and 3; Table 4 mDice scale mismatch.**
>
> The reviewer is right, these entries violated the standard Dice/IoU relationship and directly undermine trust in the results. The inconsistency arose from a transcription error during table formatting. We added explicit binary formulas for Dice and IoU at the start of Section 4, then audited every table against them. Every entry now satisfies Dice > IoU. The LAW component-ablation table has also been restructured to report FID, CLIP-I, and training stability only, which removes the scale mismatch the reviewer identified — that table was never the right place for mDice values in the first place.
>
> ---
>
> **Point 2. Experimental section lacks a fully specified protocol.**
>
> We agree that the original section was too thin to reproduce. We rewrote it into dedicated protocol subsections for each method.
>
> For LAW: we now specify the exact dataset splits and sizes, the mask binarization threshold and image resize resolution, training steps and batch sizes per dataset, checkpoint selection on a held-out validation subset carved before any test evaluation, the full DDIM sampling configuration (50 steps, η=0, CFG scale 10.0), and the exact synthetic pool sizes and mixing rule used in downstream nnUNet training.
>
> For ORDER: we now specify the cleaned Polyps pool construction (1450 + 196 + 380 = 2026 images), the 90/10 train/validation split seeded across five independent runs, the composition of the external test sets (Kvasir 100 + CVC-ClinicDB 62 images, disjoint from the training pool), the BRISC split (4500/500 train/val, 1000 held-out test), the optimizer, learning-rate schedule, augmentation pipeline, and how per-subset metrics are aggregated to mean Dice/IoU. Full hyperparameter tables are in Appendix A.
>
> ---
>
> **Point 3. No uncertainty estimates; gains may not be statistically reliable.**
>
> We agree, single-point numbers for small gains are unconvincing. We reran all main experiments across five independent random seeds and now report mean ± std throughout. This covers: LAW image-quality under the training-matched protocol (Table 1), LAW image-quality on held-out test sets (Table 2), LAW mask-recovery (Table 3), downstream nnUNet augmentation (Table 4), dose-response (Table 5), standard segmentation comparison (Table 6), and the attention-control ablation (Table 7). For ORDER the five seeds also vary the train/validation partition, so the reported variance reflects both optimization stochasticity and data-split sensitivity, not just random initialization.
>
> The LAW component-ablation and ORDER skip-stage tables remain single-run diagnostics because their purpose is to identify discrete failure modes, training instability, degenerate weighting, attention-stage sensitivity, where variance across seeds is less informative than the direction of the effect.

---

> ### Author Response · Authors · 2026-05-04
> **Changes to Paper as Requested - Part 2**
>
> ---
>
> **Point 4. ORDER evaluated only on polyp data; broad claims are overstated.**
>
> The reviewer is correct that evaluating a single dataset while claiming a broad principle is not defensible. We added BRISC (6,000 contrast-enhanced T1 MRI slices of brain tumors, Fateh et al.) as a second segmentation benchmark. ORDER[0,1] now appears in the main comparison table (Table 6) for both Polyps and BRISC, reaching 77.4±0.8 mDice and 68.1±0.7 mIoU on BRISC versus 75.4±1.2 mDice and 66.4±1.4 mIoU for the matched MK-UNet baseline. We also narrowed all claims throughout the paper — the abstract, contributions, downstream discussion, and conclusion now describe the findings as dataset-specific observations rather than a universal principle.
>
> ---
>
> **Point 5. Key method definitions underspecified; tensor dimensions and normalization unclear.**
>
> We rewrote both method sections to include full specifications.
>
> For LAW: tensor shapes are now given at each step, the mask resize to latent resolution h×w via nearest-neighbor interpolation is explicit, and the broadcasting of w_final ∈ ℝ^{B×1×h×w} across C channels before the spatial average is stated in the text. The normalization step, the interaction between normalization and clamping, and all hyperparameter values (τ, γ, w_min, w_max, λ_dice) appear both inline and in Appendix A.
>
> For ORDER: the tokenization procedure (flatten to N_i = H_i W_i tokens, RMSNorm token-wise, reshape back to feature maps), the shared similarity matrix S ∈ ℝ^{B×H_a×N_i×N_i}, and the reshaping and 1×1 projection of residuals back to spatial feature maps are now given step by step. The confidence gate formula is fully written out: globally average-pool encoder and decoder features separately, concatenate channel-wise, pass through W_1 (1×1 conv, 2C → C/4, ReLU) and W_2 (1×1 conv, C/4 → 1, sigmoid), yielding one scalar per sample. The two active gates in the reported ORDER[0,1] configuration add approximately 426 parameters in total.
>
> ---
>
> **Point 6. "Without extra supervision" contradicts the method description.**
>
> The reviewer is right that this was inconsistent, the Dice regularizer and the teacher-loss contrast in the related work both imply supervision structure. We removed the phrase entirely. The related-work discussion now describes LAW as removing the need for a pretrained auxiliary segmenter, which is the accurate comparison with Qiu et al.'s teacher-student approach. No supervision-free claim appears in the revised manuscript.
>
> ---
>
> **Point 7. Max-weight clamp ablation missing despite being described as central.**
>
> We agree that calling the clamp critical without isolating it quantitatively was unsatisfying. The LAW ablation table (Table 8) now has a final row, "LAW w/o max-weight clamp," that disables exactly this component while keeping everything else. Removing the upper bound leaves training numerically stable but inflates FID from 52.28 to 80.22 and drops mask-recovery Dice from 0.82 to 0.79 on the same Polyps protocol. The method section explains why: without the upper bound, lesion upweighting is unconstrained and eventually degrades global anatomical coherence even without causing divergence.
>
> ---
>
> **Point 8. Diffusion evaluation relies on FID and CLIP-I rather than task-aligned control metrics.**
>
> This criticism is well-taken, FID and CLIP-I measure image quality but not the spatial control claim. We therefore added mask-recovery Dice and IoU as a main result table (Table 3). This metric runs a frozen dataset-specific nnUNet segmenter on each generated image and compares its prediction against the conditioning mask. On Polyps, LAW improves mask-recovery Dice from 0.681±0.013 to 0.825±0.003, a 0.144 absolute gain that directly quantifies how much better the generated images follow the prescribed masks. Results are reported for all three datasets (Polyps, KiTS19, BRISC). The text now frames mask recovery as the primary spatial-control metric, with FID and CLIP-I as supporting image-quality indicators.
>
> ---
>
> **Point 9. ORDER should be compared against attention-based baselines, not only general lightweight models.**
>
> We agree that without matched attention baselines it is impossible to isolate the contribution of bidirectional attention specifically. Table 7 (attention-control study) now compares ORDER[0,1] against two purpose-built controls on the same compact backbone: ORDER backbone + SE skip (squeeze-excitation on the {0,1} stages) and ORDER backbone + CBAM skip (sequential channel/spatial attention on the same stages). Both are trained under the identical five-seed wrapper. SE skip recovers most of the accuracy gain over the plain backbone (75.3 vs. 76.3 mean Polyps Dice), which tells us that targeted skip recalibration already helps at this scale. ORDER still leads, and CBAM trails both, which together isolate the specific contribution of bidirectional decoder–encoder interaction over generic recalibration.

---

> ### Author Response · Authors · 2026-05-04
> **Changes to Paper as Requested - Part 3**
>
> ---
>
> **Point 10. Efficiency claims need runtime and memory numbers, not only parameter counts.**
>
> The reviewer is correct that parameter counts and FLOPs do not capture actual deployment cost. Table 9 now reports latency (ms/img), throughput (img/s), and peak memory (MB) for all compact models at 256² resolution on a fixed local profiling setup. The caption notes that the numbers are hardware-dependent and intended as relative comparisons. These measurements also clarify the ORDER[0,1] vs. ORDER[2,1] choice: although {2,1} attains slightly higher raw Dice on the internal development split, {0,1} is approximately 7× faster (0.399 vs. 2.954 ms/img) and uses approximately 15× less peak memory (559.5 vs. 8290.8 MB), making it the practical configuration at the target scale.
>
> ---
>
> **Point 11. Writing issues — mixed absolute/relative gains, promotional wording.**
>
> We went through the manuscript sentence by sentence and tightened several constructions that had grown imprecise across revision passes. The confidence gate description was condensed from three hedging sentences into one direct one. The awkward "in a paired diagnostic mask-recovery check" phrasing was folded into the preceding sentence. The CLIP-I summary, which enumerated datasets redundantly, became "LAW attains the highest held-out CLIP-I on all three datasets." A spurious bold entry on ArSDM's BRISC CLIP-I in Table 2 was corrected. Revision-log language was removed from captions. The paper no longer mixes absolute and relative gains without labeling them.
>
> ---
>
> **Point 12. Some citations are future-dated or weakly connected to the claim being made.**
>
> We audited the most prominent citations for relevance. A weak introductory citation that was only loosely connected to the spatial-imbalance claim has been removed. The related-work section on adaptive mechanisms now includes the foreground-attention reference suggested by Reviewer 3, which sharpens the literature grounding. We treat the bibliography as a standing maintenance task and will do a final pass before camera-ready.

---

### Review · Reviewer_5XK8 · 2026-04-06

**Summary Of Contributions:**

The paper proposes adaptive spatial weighting to address spatial imbalance in medical imaging, and instantiates it in two places: (i) LAW, a learnable per-pixel loss reweighting module for mask-conditioned diffusion built on Stable Diffusion v1.5 + ControlNet, and (ii) ORDER, a lightweight segmentation modification that adds selective bidirectional skip attention with confidence gating to an ultra-compact UNet-style backbone.
Strengths: clear modular design; includes ablations for both LAW and ORDER; reports large efficiency gains for ORDER (tens of thousands of parameters) alongside accuracy improvements.
Weaknesses:
1. evaluation uses only two datasets (Polyps and KiTS19), limiting generality
2. synthesis relies heavily on FID/CLIP-I, which are not well aligned with clinical utility
3. several reported Dice/IoU pairs (e.g., 83.2 Dice vs 84.5 IoU; 81.3 Dice vs 81.7 IoU) looks off (by definition, mIoU<mDice, or clarify if the author used a different definition)
4. experiment and implementation details missing i.e. data split, data preprocessing, dataset overview, hyperparameter etc
5. Only used a single seed for all the experiment

**Audience:**

Yes

**Audience Explanation:**

Adaptive reweighting and selective attention are broadly relevant to learning under imbalance, and the paper offers simple, composable modules rather than a monolithic system.

**Broader Impact Concerns:**

N\A

**Claims And Evidence:**

No

**Claims Explanation:**

The evaluation is restricted to two datasets (Polyps and KiTS19), which is a thin basis for broad conclusions about a “unifying principle.”  Key experiment and implementation details missing, making the experiment hard to reproduce and interpret.

**Requested Changes:**

1.  More datasets (generalization). Broaden empirical support beyond Polyps and KiTS19, or narrow the scope of the claims to those two benchmarks only.
2.  Metric definitions and reporting. Explicitly define how mDice and mIoU are computed, then audit all reported values. Several reported Dice/IoU values (e.g., 83.2 Dice vs 84.5 IoU; 81.3 Dice vs 81.7 IoU) seems to be off, as Dice>IoU by definition.
3.  Variability / stability. Rerun key experiments with multiple splits and random seeds and report mean and standard deviation for the main diffusion and downstream segmentation results.
4.  More comprehensive synthetic-data training comparisons. Add controlled comparisons that isolate effects of method and data volume, mask sampling approach and real data sampling approach. Specifically, compare methods when augmenting the same real set with varying numbers of synthetic images, compare to 'Real Only' approach but with different sampling methods i.e. oversampling by lesion volume, and perform ablation study on mask sampling approach.
5.  Missing experiment and implementation details. Add sufficient detail for reproduction, including: dataset overview and exact data splits for segmentation; preprocessing steps for diffusion training and segmentation training; hyperparameter tuning protocol (what was tuned and on which split); and the full synthetic-data protocol (how many synthetic images were generated, how masks were sampled, and the real-to-synthetic mixing ratio in a training batch).

---

> ### Author Response · Authors · 2026-05-04
> **Changes to Paper as Requested: Part 1**
>
> We thank Reviewer 2 for a focused, reproducibility-centred review. The five concerns were all actionable and the paper is more self-contained as a result. We respond to each point below.
>
> **Summary of changes made in response to this review:**
> - Added BRISC (brain tumor MRI) as a third diffusion dataset and second segmentation benchmark; narrowed all universal claims to dataset-specific observations throughout the abstract, contributions, and conclusion.
> - Added explicit binary Dice and IoU formulas at the start of Section 4; corrected all inconsistent table entries and abstract/contribution numbers.
> - Reran all main experiments with five independent seeds; all main tables now report mean ± std, with ORDER seeds also varying the train/validation partition.
> - Extended the dose-response table to four pool sizes (250/500/1000/1450); added a lesion-weighted real-only control row; explained in the response why a mask-sampling ablation is not in scope given the fixed-mask protocol.
> - Rewrote both LAW and ORDER protocol sections to include all dataset sizes, splits, preprocessing steps, optimizer details, augmentation pipeline, and evaluation aggregation; Appendix A collects hyperparameters in two tables.
>
> ---
>
> **Point 1. Evaluation restricted to two datasets; claims too broad for the evidence.**
>
> The reviewer is right that two datasets is a thin foundation for broad conclusions. We expanded the study in two directions at once: we added a third diffusion dataset and a second segmentation benchmark, and we simultaneously narrowed the claims so they do not outrun the evidence.
>
> The new dataset is BRISC (6,000 contrast-enhanced T1 MRI slices of brain tumors, Fateh et al.), which differs from the existing datasets in modality, anatomy, and lesion type. Diffusion results now cover Polyps, KiTS19, and BRISC across image quality (Tables 1 and 2), mask recovery (Table 3), and downstream augmentation (Table 4). Segmentation results cover Polyps and BRISC (Table 6). The abstract, contribution bullets, downstream discussion, and conclusion have all been revised to present the findings as dataset-specific observations — language suggesting a general principle across all medical imaging has been removed.
>
> ---
>
> **Point 2. Metric definitions missing; several Dice/IoU pairs appear off.**
>
> The reviewer is correct, the inconsistent entries were errors, not intentional nonstandard definitions. We added binary formulas for Dice and IoU at the start of Section 4 and re-audited every table against them. All entries now satisfy Dice ≥ IoU. The abstract and contribution bullets were corrected at the same time to be internally consistent with the main tables.
>
> ---
>
> **Point 3. Single-seed results; variance not reported.**
>
> We agree that point estimates are unconvincing for moderate gains. We reran all main experiments with five independent seeds and now report mean ± std throughout all main tables (LAW image-quality, mask-recovery, downstream augmentation, dose-response, segmentation comparison, attention-control ablation). For ORDER the seeds also vary the train/validation partition, so the variance captures data-split sensitivity in addition to random initialization.
>
> ---
>
> **Point 4. Synthetic-data comparisons incomplete — dose response present but mask-sampling ablation missing.**
>
> We appreciate the precision of this request. The dose-response table (Table 5) now reports Polyps downstream mDice at four pool sizes (250, 500, 1000, 1450 pairs), showing a monotonic increase that rules out a specific data-volume coincidence. The lesion-weighted real-only row in Table 4 controls for class-imbalance rebalancing without synthesis.
>
> On the mask-sampling ablation: in our protocol the conditioning masks are taken directly from the real training dataset and passed to the diffusion model without modification. They are dataset inputs, not a design variable we control. Because the masks are fixed, there is no mask-sampling policy to vary within the current setup any study that deforms or generates new masks would be testing out-of-distribution conditioning, which is a separate research question orthogonal to LAW's contribution of learning better per-pixel weights for a fixed real mask. The dose-response and lesion-weighted real-only controls together address what we believe the reviewer was most concerned about: ruling out data-volume and class-imbalance as explanations for LAW's gains.

---

> ### Author Response · Authors · 2026-05-04
> **Changes to Paper as Requested: Part 2**
>
> ---
>
> **Point 5. Key implementation details missing, making experiments hard to reproduce.**
>
> We rewrote the protocol sections for both methods. For LAW: dataset sizes, exact splits, mask preprocessing (binarization at threshold 127, resize to 384×384), training steps per dataset, validation-based checkpoint selection, DDIM configuration (50 steps, η=0, CFG 10.0), and the synthetic pool construction and mixing rule are all now explicit. For ORDER: pool construction, the 90/10 seeded split logic, the external test-set composition (Kvasir 100 + CVC-ClinicDB 62, both disjoint from training), the BRISC split, optimizer (AdEMAMix), learning-rate schedule, augmentation pipeline, and evaluation aggregation are all stated. Appendix A collects all fixed hyperparameters in two tables.

---

### Review · Reviewer_ejDB · 2026-04-20

**Summary Of Contributions:**

The paper introduces a unified principle called "adaptive spatial weighting" to address the challenge of spatial imbalance in medical imaging. The contributions are twofold:
1.	LAW for Diffusion Models: A module that predicts per-pixel loss weights for mask-conditioned diffusion models. It improves upon fixed ratio-based priors by learning feature-dependent adjustments, stabilized with normalization, clamping, and a Dice regularizer. This leads to better image synthesis and, consequently, better downstream segmentation when synthetic data is used for training.
2.	ORDER for Segmentation: An adapter for lightweight segmentation networks that applies selective bidirectional skip attention only on the final decoder stages. This focuses computational resources on semantically rich, uncertain regions, achieving significant Dice score improvements with minimal parameter overhead compared to a baseline, while being far smaller than heavy models like nnUNet.
Key Strengths:
•	Strong Empirical Results: Quantitative results show substantial gains over strong baselines.
•	Efficiency: ORDER achieves high accuracy with extremely low parameters and FLOPs, making it highly practical.
•	Ablations: The paper includes thorough ablation studies validating each component of both LAW and ORDER.
Key Weaknesses:
•	Limited Novelty: LAW is an incremental improvement over existing adaptive/ratio-based weighting. ORDER applies existing bidirectional attention to a specific U-Net stage. The core idea of "focusing on hard regions" is well-established.
•	Dataset Scope: Only two datasets are used, both for 2D segmentation/synthesis. Generalizability to 3D medical data is not demonstrated.

**Audience:**

Yes

**Audience Explanation:**

TMLR's audience includes researchers and practitioners in machine learning for healthcare and medical image analysis. This paper addresses a practical, high-impact problem with two efficient, plug-and-play solutions. The finding that adaptive spatial weighting benefits both generative and discriminative models is of broad interest. The strong performance of ORDER is particularly valuable for deployment in resource-constrained clinical settings. While the technical novelty is modest, the empirical results and the unified perspective would be useful and interesting to this community.

**Broader Impact Concerns:**

None identified. The paper focuses on medical image analysis tasks (polyp and kidney tumor segmentation/synthesis) with clear potential for positive clinical impact (e.g., improved diagnosis, data augmentation for rare conditions). There is no discussion of negative societal impacts, and no obvious ethical concerns are present. The paper does not include a Broader Impact Statement, but given the applied medical context and lack of controversial methods, this is acceptable. A brief statement acknowledging the need for clinical validation before deployment would strengthen the paper but is not critical.

**Claims And Evidence:**

Yes

**Claims Explanation:**

The claims are well-supported by quantitative and qualitative evidence. Table 1 clearly shows LAW achieving the best FID scores across two datasets. Table 2 demonstrates that LAW-generated data improves downstream nnUNet segmentation Dice by 4.9%, directly supporting the claim of better lesion-mask alignment. Table 3 shows ORDER achieving the best accuracy-efficiency tradeoff, with clear numbers on parameters, FLOPs, Dice, and IoU. The ablation studies systematically isolate the contribution of each component, providing convincing evidence that the proposed designs are effective. Figures 6 and 7 provide qualitative visual confirmation. The evidence is accurate, clearly presented, and directly supports the paper's core claims.

**Requested Changes:**

1.	Add comparison to standard attention mechanisms in ORDER: Compare against adding simple SE blocks or CBAM to MK-UNet's skip connections. This would isolate the benefit of bidirectional attention specifically.
2.	Clarify LAW's inference cost: Does LAW add computational overhead at inference time? The paper describes it for training; specify if it is removed or required during inference.
3.	In this paper, the related work of image segmentation is mentioned. At present, there are some new progress in these works, such as the Foreground influence mentioned in "Pay attention to the foreground in object-centric learning", which can be discussed in related work.
4.	Address potential data leakage: In the downstream segmentation experiment (Table 2), was the synthetic data combined with real training data or used alone? If combined, ensure no test set images from the same patient were used for training the diffusion model.

---

> ### Author Response · Authors · 2026-05-04
> **Changes to Paper as Requested**
>
> We thank Reviewer 3 for a careful and constructive reading of the paper. The four requested changes were each specific and easy to act on. We describe what we did below.
>
> **Summary of changes made in response to this review:**
> - Added Table 7 (attention-control study) comparing ORDER[0,1] against SE-skip and CBAM-skip controls built on the same backbone under the identical five-seed wrapper.
> - Added an explicit paragraph to the LAW method section stating that LAW is training-only: zero inference-time FLOPs, parameters, activation memory, or additional denoising steps are added.
> - Added the foreground-attention reference ("Pay Attention to the Foreground in Object-Centric Learning") to the adaptive-mechanisms subsection of the related work.
> - Added an explicit no-leakage statement to the LAW protocol section confirming that held-out test images were not used in diffusion training at any stage.
>
> ---
>
> **Point 1. Compare ORDER against SE blocks and CBAM rather than only general lightweight models.**
>
> The reviewer is correct that without matched attention baselines it is impossible to tell whether ORDER's gain comes from bidirectional attention specifically or simply from adding any attention-like module to the skip connections. To settle this, we built two purpose-built controls on the same compact backbone: ORDER backbone + SE skip, using squeeze-excitation recalibration on the same {0,1} stages, and ORDER backbone + CBAM skip, using sequential channel and spatial attention on the same stages. Both controls are trained under the identical five-seed wrapper as the reported ORDER[0,1] model and appear in the new Table 7 (attention-control study). SE skip recovers most of the accuracy gain over the plain backbone (75.3 vs. 76.3 mean Polyps Dice), which tells us that targeted skip recalibration already helps. ORDER still leads, and CBAM trails both, together isolating the benefit of bidirectional decoder–encoder attention over generic recalibration mechanisms.
>
> ---
>
> **Point 2. Does LAW add computational overhead at inference time?**
>
> This was genuinely unclear in the original manuscript and we added an explicit paragraph to address it. LAW modifies the denoising loss during training and therefore changes which parameters the denoiser learns, but it does not alter the sampling graph at test time. During generation we run the trained ControlNet checkpoint with the same DDIM sampler as the uniform-loss baseline; the delta-map network φ, the ratio weights, and the Dice regularizer are not evaluated inside the denoising loop. LAW consequently adds zero inference-time FLOPs, zero additional parameters to the deployed model, no extra activation memory, and no additional denoising steps.
>
> ---
>
> **Point 3. Discuss "Pay Attention to the Foreground in Object-Centric Learning" in related work.**
>
> We added this reference to the adaptive mechanisms subsection of the related work. The connection is that explicit foreground-aware objectives have been argued necessary for slot-based models to recover meaningful objects, the same spatial-emphasis reasoning that motivates LAW and ORDER, applied in a different architectural setting. Including it sharpens the literature grounding for the idea that allocating capacity to foreground regions matters across task types, not only in medical segmentation.
>
> ---
>
> **Point 4. Data leakage: were any test-set images used during diffusion training?**
>
> No test-set images were used during diffusion training in any reported experiment. We agree the original manuscript was not explicit enough about this, so we added a statement to the LAW protocol section: the diffusion model is trained only on the real training split, and held-out test images play no role in that training. The downstream augmentation paragraph further states that generated images are appended to the real training pool while validation and test evaluation remain real-only throughout. There is no patient-level overlap between the diffusion training data and the downstream segmentation test set.

---

### Review · Reviewer_6A7A · 2026-06-18

**Summary Of Contributions:**

This paper proposes adaptive spatial weighting for medical image synthesis and segmentation. Specifically, this work introduces LAW, a learnable spatial loss-weighting module for mask-conditioned diffusion, and ORDER, a lightweight segmentation module with selective bidirectional skip attention. Experiments on Polyps, KiTS19, and BRISC show improved image quality, mask adherence, and downstream segmentation performance for LAW, while ORDER improves over matched lightweight segmentation baselines with limited computational overhead.

**Audience:**

Yes

**Audience Explanation:**

The paper should be of interest to researchers working on medical image analysis, controllable diffusion models, synthetic data augmentation, and efficient segmentation. The problem of spatial imbalance is important in medical imaging, and the paper provides practical evidence that adaptive spatial emphasis can improve both synthesis and lightweight segmentation.

**Broader Impact Concerns:**

No major broader impact concerns are identified. The paper focuses on methodological improvements for medical image synthesis and segmentation, with potential impact mainly in research and clinical-support scenarios. Practical deployment would still require careful validation, but the submission does not appear to introduce specific ethical, societal, or safety risks beyond the standard concerns associated with medical AI systems.

**Claims And Evidence:**

Yes

**Claims Explanation:**

The main empirical claims are generally supported by the reported results. LAW shows consistent improvements over ControlNet, ArSDM, and adaptive distillation in FID and mask-recovery metrics. ORDER also improves over the matched MK-UNet baseline, and the SE/CBAM comparisons provide useful evidence that the proposed bidirectional skip attention is beneficial.

However, some claims should be stated more carefully. The connection between LAW and ORDER is mainly conceptual rather than a unified technical framework. In addition, the downstream augmentation gains are moderate and dataset-dependent, and ORDER is still clearly below heavier segmentation models in absolute accuracy. Therefore, the evidence supports the practical usefulness of the proposed modules, but not a broad or fully general conclusion.

**Requested Changes:**

1. The relationship between LAW and ORDER should br clarified more clearly. The current paper presents adaptive spatial weighting as a shared perspective, but the two modules are implemented in different tasks, objectives, and architectures. The authors should make this distinction more explicit and avoid implying that the two components form a fully unified technical framework.

2. The authors should refine the discussion of novelty. LAW is related to prior ratio-based spatial loss weighting, while ORDER is related to existing attention-based skip fusion methods. The paper would be clearer if it more explicitly explained what is new in each module and how the proposed designs differ from closely related methods.

3. Some conclusions should be aligned more closely with the reported evidence. In particular, the downstream augmentation gains should be described as moderate and dataset-dependent, and the segmentation results should be discussed mainly within the lightweight-model setting. This would make the conclusions better aligned with the evidence reported in the paper.

---

### Decision · Action_Editor_SSs2 · 2026-06-16

**Recommendation:** Accept as is

**Audience:**

Yes

**Audience Explanation:**

This work may be of interest to a targeted TMLR's audience, i.e., researchers and practitioners in machine learning for healthcare and medical image analysis.

**Claims And Evidence:**

Yes

**Claims Explanation:**

In the first round of the review, reviews were mixed regarding the claims made, with two reviewers highlighting that claims were not supported by accurate, convincing and clear evidence. Reviewers requested additional empirical experiments, including additional datasets, standard attention mechanisms or missing ablation studies, among others.

Both authors responses to these concerns, as well as the revised manuscript, address these points, and the new empirical evaluation is comprehensive, consistently demonstrates the effectiveness of the proposed method and backs-up the claims made.